# A composition-dependent molecular clutch between T cell signaling condensates and actin

**Jonathon A Ditlev[1,2†], Anthony R Vega[3†], Darius Vasco Köster[1,4†‡], Xiaolei Su[1,5§], Tomomi Tani[6], Ashley M Lakoduk[7], Ronald D Vale[1,5], Satyajit Mayor[1,4*], Khuloud Jaqaman[3,8*], Michael K Rosen[1,2*]**

[1]Howard Hughes Medical Institute, Summer Institute, Marine Biological Laboratory, Woods Hole, United States; [2]Department of Biophysics, Howard Hughes Medical Institute, University of Texas Southwestern Medical Center, Dallas, United States; [3]Department of Biophysics, University of Texas Southwestern Medical Center, Dallas, United States; [4]National Centre for Biological Sciences, Tata Institute for Fundamental Research, Bangalore, India; [5]Department of Cellular and Molecular Pharmacology, Howard Hughes Medical Institute, University of California, San Francisco, San Francisco, United States; [6]Eugene Bell Center for Regenerative Biology and Tissue Engineering, Marine Biological Laboratory, Woods Hole, United States; [7]Department of Cell Biology, University of Texas Southwestern Medical Center, Dallas, United States; [8]Lyda Hill Department of Bioinformatics, University of Texas Southwestern Medical Center, Dallas, United States

*For correspondence:
mayor@ncbs.res.in (SM);
khuloud.jaqaman@
utsouthwestern.edu (KJ);
michael.rosen@utsouthwestern.
edu (MKR)

†These authors contributed equally to this work

Present address: ‡Centre for Mechanochemical Cell Biology and Division of Biomedical Sciences, Warwick Medical School, University of Warwick, Coventry, United Kingdom; §Department of Cell Biology, Yale School of Medicine, New Haven, United States

Competing interests: The authors declare that no competing interests exist.

**Abstract** During T cell activation, biomolecular condensates form at the immunological synapse (IS) through multivalency-driven phase separation of LAT, Grb2, Sos1, SLP-76, Nck, and WASP. These condensates move radially at the IS, traversing successive radially-oriented and concentric actin networks. To understand this movement, we biochemically reconstituted LAT condensates with actomyosin filaments. We found that basic regions of Nck and N-WASP/WASP promote association and co-movement of LAT condensates with actin, indicating conversion of weak individual affinities to high collective affinity upon phase separation. Condensates lacking these components were propelled differently, without strong actin adhesion. In cells, LAT condensates lost Nck as radial actin transitioned to the concentric network, and engineered condensates constitutively binding actin moved aberrantly. Our data show that Nck and WASP form a clutch between LAT condensates and actin in vitro and suggest that compositional changes may enable condensate movement by distinct actin networks in different regions of the IS.
DOI: https://doi.org/10.7554/eLife.42695.001

## Introduction

Biomolecular condensates are compartments in eukaryotic cells that concentrate macromolecules without an encapsulating membrane (*Banani et al., 2017*; *Shin and Brangwynne, 2017*). Numerous condensates are found in the cytoplasm and nucleoplasm, where they are involved in processes ranging from mRNA storage and degradation to DNA repair and ribosome biogenesis (*Brangwynne et al., 2011*; *Feric et al., 2016*; *Luo et al., 2018*; *Protter and Parker, 2016*). They are also found at membranes, where they control the organization, and likely the activity, of many signaling receptors (*Banjade and Rosen, 2014*; *Case et al., 2019*; *Su et al., 2016*; *Zeng et al., 2018*).

Condensates are thought to form through phase separation driven by multivalent interactions between molecules containing multiple binding elements (*Banani et al., 2017*; *Banjade and Rosen, 2014*; *Li et al., 2012*). Recent models suggest that a limited collection of proteins and/or RNA molecules forms the essential phase separating scaffold of particular condensates (*Banani et al., 2016*; *Ditlev et al., 2018*; *Langdon et al., 2018*). These molecules then recruit larger numbers of client proteins to complete the structure. Condensate composition is thus determined by the specificity of interactions among scaffolds and between scaffolds and clients. The concentrations, interactions, and dynamics of scaffolds and clients are believed to dictate the biochemical activities, and consequent cellular functions, of individual condensates (*Banani et al., 2017*; *Holehouse and Pappu, 2018*; *Stroberg and Schnell, 2018*). The composition of many condensates is known to change in response to signals, implying regulated changes in activity (*Chen et al., 2008*; *Dellaire et al., 2006*; *Markmiller et al., 2018*; *Salsman et al., 2017*; *Youn et al., 2018*). However, the relationships between composition and biochemical/cellular functions are not well understood in most cases.

During activation of a T cell by an antigen presenting cell (APC), condensates organized around the transmembrane adaptor protein LAT (Linker for Activation of T cells) play important roles in downstream signaling and resultant T cell activation (*Balagopalan et al., 2013*; *Houtman et al., 2006*; *Kumari et al., 2015*). These condensates are located at the interface between the T cell and APC, known as the immunological synapse (IS) (*Bunnell et al., 2002*). We recently provided evidence that these structures form through multivalency-driven phase separation of LAT and its intracellular binding partners Grb2, Sos1, SLP-76, Nck, and WASP (*Su et al., 2016*). In this system LAT, Grb2, and Sos1 appear to act as the key scaffolds, which recruit SLP-76, Nck, and WASP as clients. During T cell activation, LAT condensates that initially appear in the periphery of the IS are moved radially to the IS center by two different actin cytoskeletal networks, a peripheral dendritic meshwork and more central circular arcs (*DeMond et al., 2008*; *Kaizuka et al., 2007*; *Mossman et al., 2005*; *Murugesan et al., 2016*; *Yu et al., 2010*). This translocation is essential for proper T cell responses to antigen presenting cells (*Babich et al., 2012*; *Ilani et al., 2009*; *Kumari et al., 2012*; *Yi et al., 2012*; *Yu et al., 2012*).

It has remained unknown how LAT condensates engage actin to move across the IS and whether their composition plays a role in this process (*Hammer et al., 2019*). To address these questions, we analyzed interactions between LAT condensates and actin in reconstituted biochemical systems and in cells using quantitative fluorescence microscopy and statistical analysis. Our in vitro studies revealed that LAT condensates bind actin filaments in a composition-dependent fashion, primarily through interactions of basic regions in Nck and N-WASP/WASP. In cells, we observed that Nck dissipates from LAT condensates as they traverse the IS. This loss spatially parallels the change in actin architecture from dendritic network to circular arcs. LAT condensates containing a mutant Grb2, which caused them to constitutively bind actin filaments independently of Nck, exhibited aberrant movement across the IS. These data show that Nck and N-WASP/WASP can form a molecular clutch between LAT condensates and actin in vitro. The combined cellular and biochemical data suggest a model in which LAT condensates engage actin differently depending on the density of Nck and WASP proteins in them, such that switching between compositions and actin-binding modes enables them to move radially via the two actin networks at the IS.

## Results

### LAT condensates reconstituted in vitro within actin networks move in a composition-dependent manner

We previously reconstituted LAT condensates on supported lipid bilayers (SLBs) through addition of various T cell signaling proteins to membrane-attached phospho-LAT (pLAT) (*Su et al., 2016*). In separate work, we also reconstituted membrane-associated contractile actomyosin networks by attaching actin to SLBs via the membrane-anchored actin binding domain of ezrin (eABD) in the presence of myosin II and capping protein (*Köster et al., 2016*). To examine interactions of LAT condensates with actin networks, here we combined these two systems into a single assay (*Figure 1A*). We attached polyhistidine-tagged phospho-LAT (pLAT) and polyhistidine-tagged eABD to Ni-NTA functionalized lipids within the SLB. We induced LAT phase separation into condensates by adding an increasing subset of binding partners, in the order Grb2, Sos1, phospho-SLP-76 (pSLP-76), Nck,

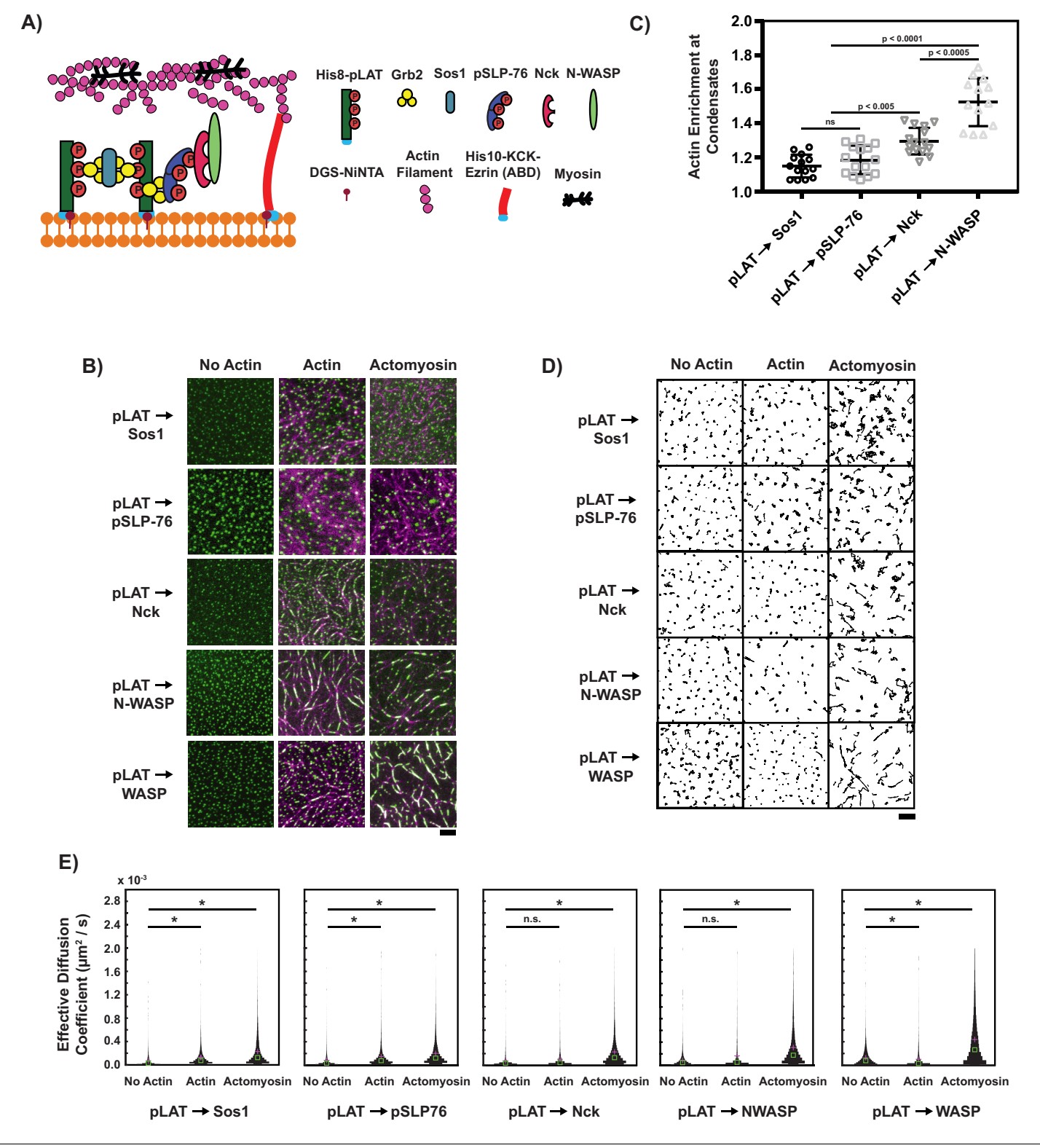

**Figure 1.** Composition regulates condensate movement with actin networks. (**A**) Schematic of biochemical reconstitution in solutions containing LAT and eABD attached to SLBs and LAT phase separation agents, actin filaments, and myosin filaments in solution. (**B**) TIRF microscopy images of LAT condensates of the indicated compositions (see text for nomenclature) without actin (left column), with actin-only networks (middle column), or with actomyosin networks (right column). pLAT-Alexa488 is green, rhodamine-actin is magenta. Scale bar = 5 μm. (**C**) Actin enrichment at condensates in reconstitution assays using actomyosin, calculated as the ratio of actin fluorescence intensity within condensates to actin intensity outside condensates

*Figure 1 continued on next page*

*Figure 1 continued*

(see Materials and methods). Shown are the individual data points and their mean ±s.d. from N = 15 fields of view from three independent experiments (with 5 FOV per experiment). P-values are for indicated distribution comparisons via Wilcoxon rank-sum test with Bonferroni correction to achieve a total type-I error of 0.05. (D) Example plots of condensate tracks. Scale bar = 5 µm. (E) Effective diffusion coefficients of the condensate tracks. Each measurement is shown as a violin plot of the distribution of values from analyzing N = 15 fields of view from three independent experiments (with 5 FOV per experiment). Green square shows median and magenta plus sign shows mean. Significance was determined by averaging results from 100 Wilcoxon rank-sum tests that compared pairs of 500 randomly-selected tracks.

DOI: https://doi.org/10.7554/eLife.42695.002

The following source data and figure supplement are available for figure 1:

**Source data 1.** Source data file for *Figure 1C*.
DOI: https://doi.org/10.7554/eLife.42695.004
**Figure supplement 1.** Detection process for LAT condensates in vitro.
DOI: https://doi.org/10.7554/eLife.42695.003

and, finally, WASP or N-WASP, as previously described (*Su et al., 2016*). Hereafter we use the nomenclature pLAT → X to indicate condensates containing pLAT and all binding partners up to X (e.g. if X is Nck, then the condensates would contain pLAT, Grb2, Sos1, pSLP-76, and Nck). In T cells, the main WASP family protein at the IS is WASP (*Kumari et al., 2014*), which acts as a constitutive complex with WASP Interacting Protein (WIP) (*Antón et al., 2002*; *Ramesh et al., 1997*). We performed experiments here with either WASP or N-WASP to examine how differences between the proteins, which are 48% identical in amino acid sequence, affect interactions of LAT condensates

with dynamic actin networks. We fused the N-WASP/WASP binding fragment of WIP to the N-terminus of each protein in order to stabilize its EVH1 domain (*Peterson et al., 2007*; *Volkman et al., 2002*), and improve bacterial expression. Our use of full-length WASP proteins here represents a step closer to the

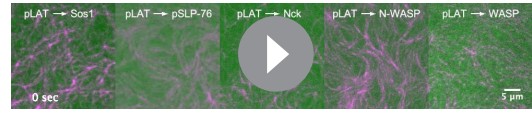

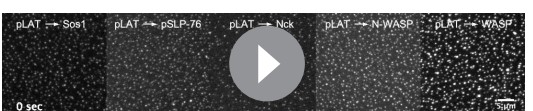

**Video 1.** Reconstitution of LAT condensate formation and movement using different component mixtures. TIRF microscopy revealed LAT condensate formation and movement on supported lipid bilayers. His$_8$-pLAT-Alexa488 was attached to Ni-functionalized SLBs at 500 molecules / µm$^2$. Condensate component mixtures are indicated at the top of each movie panel. All reconstitutions were performed in 75 mM KCl. Condensates were formed by adding 125 nM Grb2 and 125 nM Sos1 (pLAT → Sos1); 125 nM Grb2, 62.5 nM Sos1, and 62.5 nM pSLP-76 (pLAT → pSLP-76); 125 nM Grb2, 62.5 nM Sos1, 62.5 nM pSLP-76, a 125 nM Nck (pLAT → Nck); 125 nM Grb2, 62.5 nM Sos1, 62.5 nM pSLP-76, 125 nM Nck, and 125 nM N-WASP (pLAT → N-WASP); or 125 nM Grb2, 62.5 nM Sos1, 62.5 nM pSLP-76, 125 nM Nck, and 125 nM WASP (pLAT → WASP). After formation, most condensates were either immobile or displayed confined movement regardless of their composition. Movie shows a 32 µm x 32 µm field of view for each movie panel. The movie is played at nine fps with frame intervals of 15 s.

DOI: https://doi.org/10.7554/eLife.42695.005

**Video 2.** Reconstitution of LAT condensate formation and movement using different component mixtures in actin networks. TIRF microscopy revealed LAT condensate formation and movement on supported lipid bilayers in an actin network. His$_8$-pLAT-Alexa488 (green) was attached to Ni-functionalized SLBs at 500 molecules / µm$^2$. Actin filaments (magenta) were attached to the same bilayers via His$_{10}$-ezrin actin binding domains. Condensate component mixtures are indicated at the top of each movie panel. All reconstitutions were performed in 75 mM KCl. Condensates were formed by adding 125 nM Grb2 and 125 nM Sos1 (pLAT → Sos1); 125 nM Grb2, 62.5 nM Sos1, and 62.5 nM pSLP-76 (pLAT → pSLP-76); 125 nM Grb2, 62.5 nM Sos1, 62.5 nM pSLP-76, and 125 nM Nck (pLAT → Nck); 125 nM Grb2, 62.5 nM Sos1, 62.5 nM pSLP-76, 125 nM Nck, and 125 nM N-WASP (pLAT → N-WASP); or 125 nM Grb2, 62.5 nM Sos1, 62.5 nM pSLP-76, 125 nM Nck, and 125 nM WASP (pLAT → WASP). After formation, pLAT → Sos1 and pLAT → pSLP-76 condensates displayed increased mobility (compared to condensates formed in the absence of actin filaments) while pLAT → Nck and pLAT → N-WASP showed decreased mobility (compared to condensates formed in the absence of actin networks) while binding to and wetting actin filaments. Movie shows a 32 µm x 32 µm field of view for each movie panel. The movie is played at nine fps with frame intervals of 15 s.

DOI: https://doi.org/10.7554/eLife.42695.006

natural signaling system than our previous reconstitution (*Su et al., 2016*), which used only a C-terminal fragment of N-WASP, enabling us to better capture essential features of cellular LAT condensates while still maintaining manageable complexity.

For experiments involving actin, we added polymerized actin filaments that bound to the SLB via anchored eABD. To induce actin filament movement we added muscle myosin II and ATP, as previously described (*Köster et al., 2016*). While T cells express only non-muscle myosin II, the muscle isoform is functionally similar, differing largely in making somewhat longer filaments (800 nm vs 300 nm average length under similar conditions [*Vicente-Manzanares et al., 2009*]), and is much easier to purify from tissues in biochemical quantities. Since most movement of T cell receptor condensates (which typically coincide with LAT condensates) is blocked by inhibition of myosin (*Yi et al., 2012*), more complex reconstitutions including actin filament assembly and disassembly dynamics were not warranted in our work here (*Blanchoin et al., 2000*; *Didry et al., 1998*; *Shekhar and Carlier, 2017*). Thus, our reconstituted system should retain key qualitative behaviors of T cell actomyosin, while remaining experimentally practical.

We induced LAT condensate formation without actin, with actin alone, or with active actomyosin networks, and imaged the system using total internal reflection fluorescence (TIRF) microscopy (*Figure 1B*, *Videos 1*, *2* and *3*). We immediately observed that condensates containing Nck, Nck and WASP, or Nck and N-WASP associated with and wet actin filaments in both actin and actomyosin networks. In contrast, condensates lacking these proteins remained distributed across the SLB (*Figure 1B*). As a corollary, co-localization analysis (see Supplemental Methods) showed that actin enrichment in condensates increased significantly in the presence of Nck and N-WASP (*Figure 1C*).

In all conditions, we automatically detected and tracked the condensates at the SLB for 15 min, and then measured their effective diffusion coefficients using Moment Scaling Spectrum analysis as a measure of mobility (*Jaqaman et al., 2011*; *Jaqaman et al., 2008*; *Vega et al., 2018*); see Materials and methods) (*Figure 1—figure supplement 1*). This analysis revealed that in the absence of actin, condensates exhibited a distribution of diffusion coefficients with medians in the range of $3 \times 10^{-5}$ to $8 \times 10^{-5}$ $\mu m^2$ / s (*Figure 1D,E*, *Video 1*). In the presence of actin alone (i.e. no myosin), condensate movement varied with composition. pLAT → Sos1 and pLAT → pSLP76 showed an increase in mobility (*Figure 1D,E*, *Video 2*), while pLAT → Nck, pLAT → WASP, and pLAT → N-WASP had a tendency to align with actin filaments and showed either a decrease or no change in mobility (*Figure 1D,E*, *Video 2*). Lastly, in the presence of active actomyosin networks, condensates of all compositions exhibited an overall increase in mobility (median effective diffusion coefficient range of $1 \times 10^{-4}$ to $3 \times 10^{-4}$ $\mu m^2$ / s). The increase for pLAT → Sos1 and pLAT → pSLP76 was subtle, larger for pLAT → Nck, and largest for pLAT → WASP and pLAT → N-WASP. These data suggest that condensates containing Nck or, Nck and WASP, or Nck and N-WASP, which wet filaments and show a large differential in behavior between actin alone and actomyosin conditions, can be viewed distinctly from condensates containing components only up to Sos1 or Sos1 and pSLP76, which do not wet filaments and show smaller differences between the two types of actin networks.

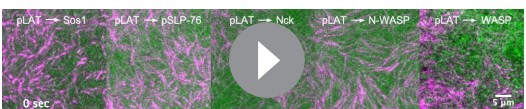

**Video 3.** Reconstitution of LAT condensate formation and movement using different component mixtures in steady-state active actomyosin networks. TIRF microscopy revealed LAT condensate formation and movement on supported lipid bilayers in an active actomyosin network. His_8-pLAT-Alexa488 (green) was attached to Ni-functionalized SLBs at 500 molecules / $\mu m^2$. Actin filaments (magenta) were attached to the same bilayers via His_10-ezrin actin binding domains. 100 nM myosin II and 1 mM ATP were added to actin filaments prior to LAT condensate formation to induce actin filament movement. Condensate component mixtures are indicated at the top of each movie panel. All reconstitutions were performed in 75 mM KCl. Condensates were formed by adding 125 nM Grb2 and 125 nM Sos1 (pLAT → Sos1); 125 nM Grb2, 62.5 nM Sos1, and 62.5 nM pSLP-76 (pLAT → pSLP-76); 125 nM Grb2, 62.5 nM Sos1, 62.5 nM pSLP-76, and 125 nM Nck (pLAT → Nck); 125 nM Grb2, 62.5 nM Sos1, 62.5 nM pSLP-76, 125 nM Nck, and 125 nM N-WASP (pLAT → N-WASP); or 125 nM Grb2, 62.5 nM Sos1, 62.5 nM pSLP-76, 125 nM Nck, and 125 nM WASP (pLAT → WASP). After formation, all condensate types displayed increased condensate mobility (compared to condensates formed in the absence of actin filaments or in an actin network). Movie shows a 32 μm x 32 μm field of view for each movie panel. The movie is played at nine fps with frame intervals of 15 s.

DOI: https://doi.org/10.7554/eLife.42695.007

## LAT condensates in vitro containing Nck and WASP or N-WASP move with contracting actomyosin networks with high fidelity

To delineate the effect of composition on the ability of LAT condensates to move with actin filaments, we devised an in vitro system where the actin filaments moved in a directional manner. This system enabled us to clearly distinguish between condensates that move with the actin filaments (because they would also exhibit directional movement) and those that do not. For this, we performed experiments where an SLB-associated actin network was induced to contract into asters by addition of myosin II filaments in the presence of low concentrations of salt and ATP (see Supplemental Methods). These experiments were especially susceptible to photo-damage caused by light exposure (*Figure 2—figure supplement 1*). Thus, imaging conditions were set to minimize light exposure and the actomyosin/condensate configuration at the end point of all time-lapses was compared to adjacent, non-imaged regions of the SLB. These actomyosin contraction experiments were performed with pLAT → Sos1, pLAT → WASP, and pLAT → N-WASP condensates. In the beginning of such experiments, pLAT → Sos1 condensates were randomly distributed on the membrane while pLAT → WASP or pLAT → N-WASP condensates were aligned along the actin filaments as observed above (*Figure 2A*). In all cases the filament network started to contract immediately upon myosin II addition and formed stable asters within 2 min (*Video 4*). As shown in *Figure 2A*, at the end of the contraction, most of the pLAT → Sos1 condensates remained scattered across the SLB, while virtually all of the pLAT → N-WASP condensates had moved with the actin into asters. pLAT → WASP condensates showed behavior intermediate between these extremes. To quantify these behaviors, we examined the speed and direction of both condensate movement and actin movement during actomyosin network contraction using Spatio-Temporal Image Correlation Spectroscopy (STICS) (*Ashdown et al., 2014*) (*Figure 2B*). We found that the speed of pLAT → N-WASP condensates correlated well with the speed of actin, while the speed of pLAT → Sos1 condensates did not (*Figure 2C*). pLAT → WASP condensates showed an intermediate degree of correlation. Additionally, the distribution of angles between the vectors of pLAT → N-WASP condensate movement and proximal actin movement showed clear preference for smaller angles, indicating a high degree of co-movement. In contrast, the angle distribution for pLAT → Sos1 showed only a slight preference for smaller angles, which was marginally significant (*Figure 2D*). Again pLAT → WASP condensates displayed an intermediate behavior. Together, the steady state (*Figure 1*) and contraction (*Figure 2*) analyses show that LAT condensates are influenced by actin network dynamics in a composition-dependent fashion. Condensates containing Nck or Nck and WASP/N WASP bind to and move with actomyosin filaments. In contrast, condensates lacking these proteins do not bind filaments appreciably, and move through a different mechanism, likely due to non-specific steric contacts. Thus, Nck and WASP/N WASP function as a molecular clutch between LAT condensates and actin in vitro.

## Phase separation on membranes enhances interactions between Nck/N-WASP and actin filaments

The data above suggest that Nck, WASP, and N-WASP mediate binding of LAT condensates to actin filaments. To test this, we quantified the recruitment of preformed actin filaments (10% rhodamine-labeled) to SLBs by LAT condensates of different compositions in the absence of eABD. As shown in *Figure 3A* and *Figure 3—figure supplement 1*, only condensates containing Nck, Nck and WASP, or Nck and N-WASP recruited substantial amounts of actin filaments. In each case, condensates were elongated along filaments, as observed above.

To better understand these interactions, we next determined the effective dissociation constants for LAT → Nck and LAT → N-WASP condensates binding to actin filaments (*Figure 3—figure supplement 2A*). We measured the fluorescence intensity of recruited actin as a function of actin filaments in solution. Fluorescence intensity showed a sigmoidal dependence on actin filament concentration, indicating cooperative recruitment to the membrane, perhaps due to avidity effects based on multi-point interactions of longer filaments with the condensates (see below). Fitting the data to a Hill equation showed that LAT → N-WASP condensates bind actin more tightly than LAT → Nck condensates ($K_D$ = 280 nM, 95% Confidence Interval (CI) [250 nM, 300 nM] versus 410 nM, 95% CI [380 nM, 450 nM]) and with similar Hill coefficients (3.6, 95% CI [3.0, 4.4] vs 3.2, 95% CI [2.4, 5.6]) (*Figure 3—figure supplement 2B,C*). LAT → N-WASP condensates also have a higher capacity for actin filaments than LAT → Nck condensates, based on a higher maximal recruited actin

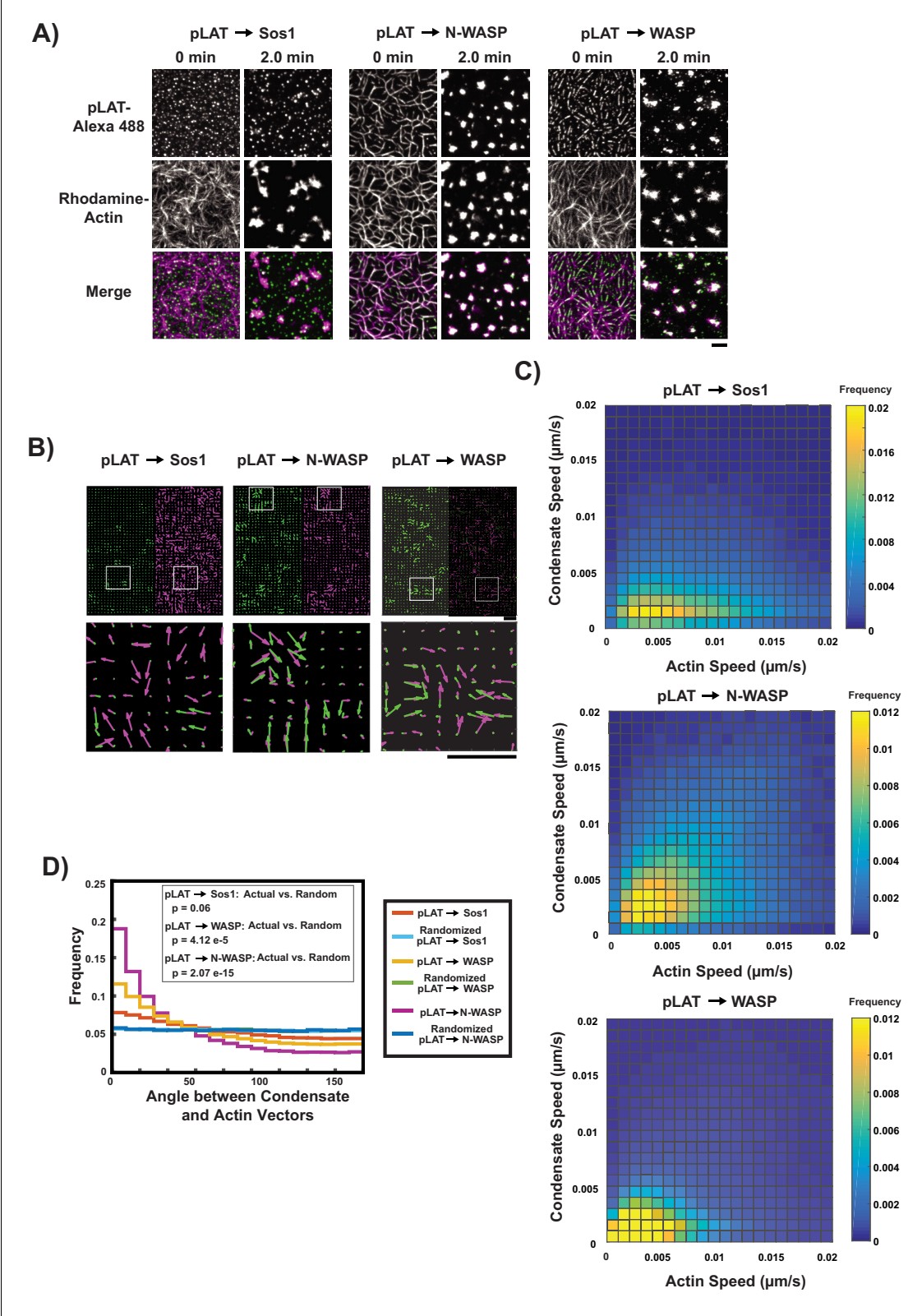

**Figure 2.** pLAT → N-WASP condensates bind to and move with moving actin filaments. (**A**) TIRF microscopy images of pLAT → Sos1 condensates (left two columns), and pLAT → N-WASP condensates (middle two columns), and pLAT→ WASP (right two columns), formed in an actin network before (t = 0 min) and after (t = 2 min) addition of myosin II. pLAT condensates are green and actin is magenta in merge. Scale bar = 5 μm. (**B–D**) STICS analysis of actin and condensate movement. (**B**) Representative map of actin (magenta) and pLAT condensate (green) vector fields. Lower panels show

*Figure 2 continued on next page*

*Figure 2 continued*

magnification of box regions in upper panels. (C) Condensate speed vs. actin speed at same position. Condensate composition indicated above each heat map. Heat map indicates frequency in each bin, that is counts in each bin normalized by total number of counts. (D) Distribution of the angle between actin and condensate movement vectors for pLAT → Sos1 (blue), pLAT → N-WASP (gold), randomized pLAT → Sos1 (red) and randomized pLAT → N-WASP (purple) (see Materials and methods for randomization). P-values are for indicated distribution comparisons via Kolmogorov-Smirnov test. Data in (C) and (D) are pooled from 15 fields of view from three independent experiments (5 FOV per experiment).

DOI: https://doi.org/10.7554/eLife.42695.008

The following figure supplement is available for figure 2:

**Figure supplement 1.** In actomyosin contractile assays, imaging conditions were chosen such that photo-damage was minimized to avoid artifactual results.

DOI: https://doi.org/10.7554/eLife.42695.009

fluorescence intensity (*Figure 3—figure supplement 2B,C*). Thus, the condensates can recruit actin filaments to membranes with high effective affinity.

We next performed co-sedimentation assays to determine whether Nck or N-WASP could bind actin filaments in solution or whether efficient binding required the proteins to be arrayed on a two-dimensional surface (*Figure 3—figure supplement 3*). At 2 μM concentration, Nck did not appreciably co-sediment with actin filaments up to 7.9 μM actin filament concentration. While these data could not yield a $K_D$ value for the interaction of Nck and actin in solution, they clearly show that binding affinity is much lower than when Nck is organized into LAT → Nck condensates on membranes. Consistent with previous reports, 2 μM N-WASP did bind filaments (*Co et al., 2007*), although only 24% of the protein co-sedimented with 7.9 μM F-actin. This contrasts with α-actinin, which has a reported $K_D$ for actin filaments of 4.7 μM (*Wachsstock et al., 1993*) and shows an approximately equal distribution between supernatant and pellet (*Figure 3—figure supplement 3*). Estimating $K_D$ of N-WASP:F-actin binding in solution from the 7.9 μM F-actin data point yields an approximate value of 25 μM. Thus, as for Nck, N-WASP has appreciably lower affinity for actin filaments in solution than when organized into LAT → N-WASP condensates on membranes.

This difference in N-WASP-actin affinity between the different formats could be due to A) effects of the other proteins in LAT → N-WASP condensates (e.g. inducing conformational changes in N-WASP that increase affinity) or B) effects of concentrating N-WASP into high-density condensates on the two-dimensional membrane surface. To test whether the collection of molecules within LAT condensates alone could increase affinity of N-WASP for actin, we performed co-sedimentation assays in which the composition of the solution was increased in complexity by adding Nck, pSLP-76, Grb2, and pLAT to N-WASP and actin filaments. In solution, addition of LAT condensate components did not enhance binding of N-WASP to actin filaments (*Figure 3—figure supplement 4*). These data suggest that it is the localization of N-WASP to LAT condensates on membranes, rather than conformation changes induced by ligands per se, that enables efficient actin filament binding. We do not know the exact source of these effects. They could arise from avidity when multiple N-WASP molecules in a high-density membrane condensate bind an actin filament. Alternatively, they could arise from actions of the membrane itself on the conformation of Nck-bound N-WASP.

Together, these data show that assembling Nck and N-WASP into phase separated condensates on membranes increases their apparent affinity for actin filaments, enabling efficient recruitment of actin filaments to membranes. This effect probably also accounts for the strong engagement of membrane-associated actin filament networks with LAT condensates containing Nck and N-WASP (*Figures 1–3*).

## Basic regions of Nck and N-WASP couple LAT condensates to actin filaments

To identify the elements of Nck that mediate LAT condensate binding to actin filaments, we attached polyhistidine-tagged fragments of the proteins to SLBs and measured their ability to recruit actin filaments from solution. Nck is composed of three SH3 domains and an SH2 domain, connected by flexible linkers of 25–42 residues (*Banjade et al., 2015*). Of these seven elements, two contain dense basic patches, one contains dense acidic patches, and the remaining four are relatively free of dense charge patches. As detailed in *Figure 3—figure supplement 5*, Nck fragments containing an excess of basic elements recruited actin to the membrane, where greater excess resulted

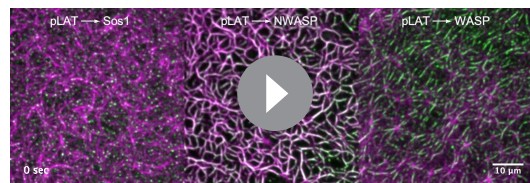

**Video 4.** Reconstitution of LAT condensate movement using different component mixtures in contracting actomyosin networks. TIRF microscopy revealed actin filament and LAT condensate movement on supported lipid bilayers in a contracting actomyosin network. His$_8$-pLAT-Alexa488 (green) was attached to Ni-functionalized SLBs at 500 molecules / $\mu m^2$. Actin filaments (magenta) were attached to the same bilayers via His$_{10}$-ezrin actin binding domains. Condensate component mixtures are indicated at the top of each movie panel. All reconstitutions were performed in 50 mM KCl. Condensates were formed by adding 125 nM Grb2 and 125 nM Sos1 (pLAT → Sos1); 125 nM Grb2, 62.5 nM Sos1, 62.5 nM pSLP-76, 125 nM Nck, and 125 nM N-WASP (pLAT → N-WASP); or 125 nM Grb2, 62.5 nM Sos1, 62.5 nM pSLP-76, 125 nM Nck, and 125 nM WASP (pLAT → WASP). After formation, pLAT → Sos1 condensates were randomly distributed within the actin network while pLAT → N-WASP condensates bound to and wet actin filaments. Actin filament contraction was induced by adding 100 nM myosin II. pLAT→ Sos1 condensates did not efficiently move with contracting actin filaments while pLAT → N-WASP condensates efficiently moved with contracting actin filaments. pLAT → WASP condensates moved with contracting actin filaments with an efficiency between pLAT → Sos1 and pLAT → N-WASP condensates. Movie shows a 52 $\mu m$ x 52 $\mu m$ field of view for each movie panel. The movie is played at five fps with frame intervals of 5 s.
DOI: https://doi.org/10.7554/eLife.42695.010

in more efficient recruitment, while neutral or acidic fragments did not. Similarly, mutating one of the basic elements (the linker between the first and second SH3 domains, L1, to neutralize it (Nck$_{Neutral}$) or to make it acidic (Nck$_{Acidic}$) greatly impaired actin recruitment, while making it more basic (Nck$_{Basic}$) enhanced actin recruitment (*Figure 3B*, *Figure 3—figure supplement 6*). These data indicate that basic regions of Nck likely contribute to binding of LAT condensates to actin filaments.

Like Nck, N-WASP also has a central basic region, amino acid residues 186–200. We thus asked whether this region and/or the two C-terminal WH2 motifs, which are known to bind actin monomers and filament barbed ends (*Bieling et al., 2018*; *Co et al., 2007*), contribute to coupling of LAT condensates to actin filaments. We generated pLAT → N-WASP condensates with N-WASP fragments consisting of the basic-proline elements (BP), basic-proline +VCA (BPVCA) elements, and a BPVCA protein with mutations in the WH2 motifs in the VCA region that impair filament binding (BPVCA$_{mut}$) (*Co et al., 2007*). All three types of condensates strongly recruited actin, indicating that WH2-actin interactions are not needed for actin filament recruitment by LAT condensates (*Figure 3—figure supplement 7*). To further examine the basic region, we generated three variants of His$_6$-tagged full length N-WASP (fused N-terminally to the WASP-binding region of WIP, as in the earlier experiments; *Figure 3—figure supplement 7*): one containing a doubled basic region (N-WASP$_{Basic}$), one wild-type (N-WASP$_{WT}$), and one containing a neutral linker instead of the basic region (N-WASP$_{Neutral}$). As shown in *Figure 3C* and *Figure 3—figure supplement 8*, these variants recruited actin in the order N-WASP$_{Basic}$ >>N WASP$_{WT}$>N WASP$_{Neutral}$. Thus, for both Nck and N-WASP, the degree of positive charge in basic elements correlates with the ability of the proteins and their LAT condensates to recruit actin filaments to SLBs.

To test whether these basic regions are necessary to couple LAT condensate movement to actin movement (*Figure 2*), we performed actin contraction assays with condensates containing N-WASP$_{Basic}$ or N-WASP$_{Neutral}$. Before myosin II addition, pLAT - > N WASP$_{Basic}$ condensates aligned almost perfectly with actin filaments, while pLAT - > N WASP$_{Neutral}$ condensates aligned only partially, consistent with the notion that the basic region mediates binding to actin filaments (*Figure 3D*). After actin network contraction, pLAT → N-WASP$_{Basic}$ condensates localized with actin asters to a similar degree as pLAT → N-WASP$_{WT}$, while pLAT → N-WASP$_{Neutral}$ localized less with actin (*Figure 3D* vs. *Figure 2A*, *Video 5* vs. *Video 4*). STICS analysis revealed that the correlation of pLAT → N-WASP$_{Basic}$ movement with local actin movement was slightly better than that of pLAT → N-WASP$_{WT}$, while the correlation of pLAT → N-WASP$_{Neutral}$ was worse (*Figure 3E,F*). The remaining correlation for pLAT → N-WASP$_{Neutral}$ is most likely due to the presence of Nck, which also contributes to actin binding.

Together these data demonstrate that regions of Nck and N-WASP that contain dense basic patches can mediate the clutch-like behaviors of the proteins by directly interacting with actin

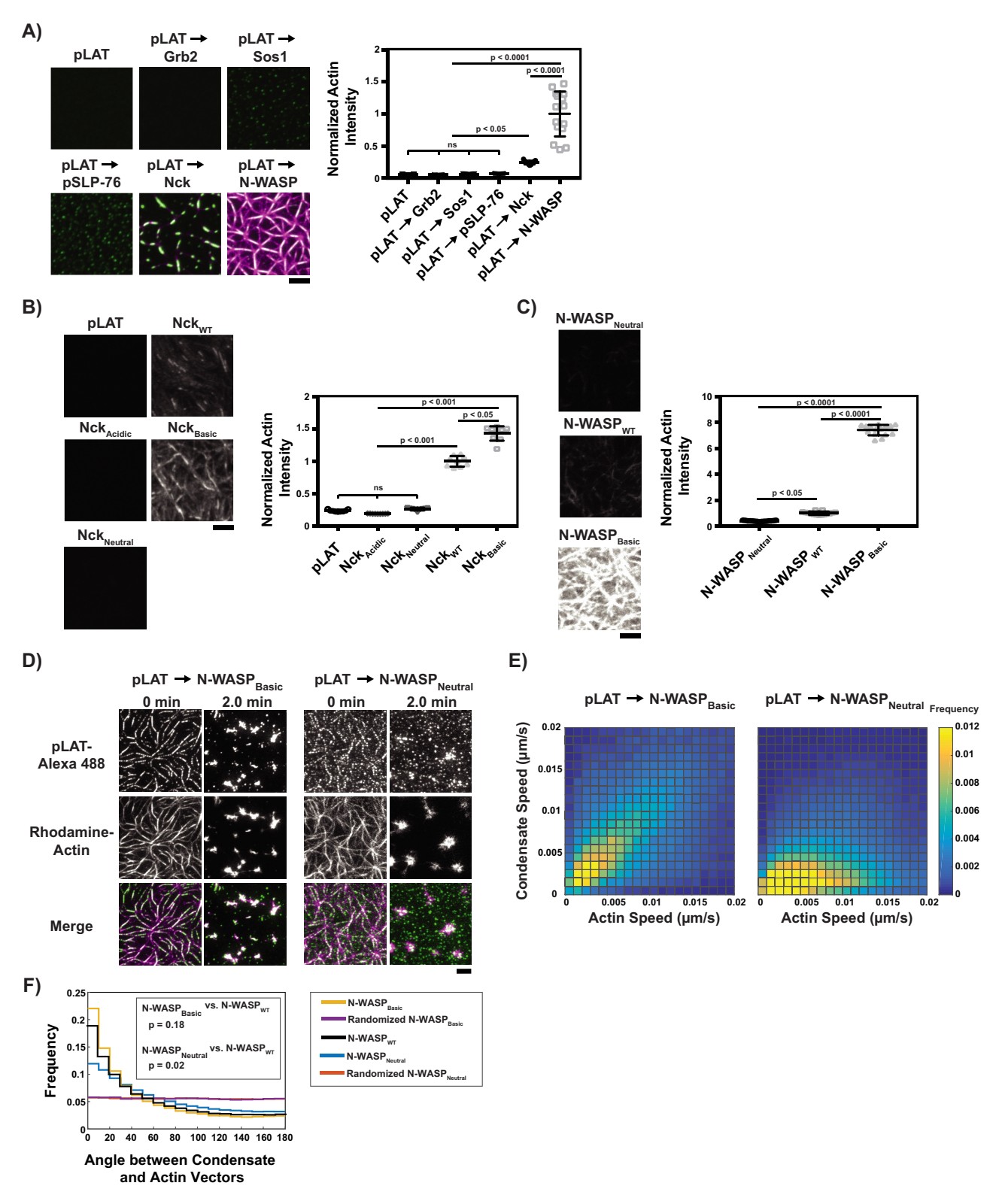

**Figure 3.** Basic regions of Nck and N-WASP mediate interaction of LAT condensates with actin filaments. (**A**) (Left) TIRF microscopy images of rhodamine-actin (magenta) recruited to SLBs by the indicated LAT condensate compositions (green). Scale bar = 5 μm. All panels use the same intensity range. (Right) Normalized average rhodamine-actin fluorescence intensity on SLBs. Shown are the individual data points and their mean ±s.d. N = 15 fields of view from three independent experiments (5 FOV per experiment). P values were determined using a t-test. (**B**) (Left) TIRF microscopy

*Figure 3 continued on next page*

*Figure 3 continued*

images of rhodamine-actin recruited to SLBs by His-tagged Nck variants. Scale bar = 5 μm. All panels use the same intensity range. (Right) Normalized average rhodamine-actin fluorescence intensity on SLBs. Shown are the individual data points and their mean ±s.d. N = 9 fields of view from three independent experiments (3 FOV per experiment). P values were determined using a t-test. (C) (Left) TIRF microscopy images of rhodamine-actin recruited to SLBs by His-tagged N-WASP variants. All panels use the same intensity range. (Right) Normalized average rhodamine-actin fluorescence intensity on SLBs. Shown are the individual data points and their mean ±s.d. N = 15 fields of view from three independent experiments (5 FOV per experiment). P values were determined using a t-test. (D) TIRF microscopy images of pLAT → N-WASP$_{Basic}$ condensates (left two columns) and pLAT → N-WASP$_{Neutral}$ condensates (right two columns) formed in an actin network before (t = 0 min) and after (t = 2 min) addition of myosin II. LAT condensates are green and actin is magenta in merge. Scale bar = 5 μm. (E) Condensate speed vs. actin speed at same position from STICS analysis. Condensate composition indicated above each heat map. Heat map indicates frequency in each bin (as in *Figure 2C*). N = 15 fields of view from three independent experiments (5 FOV per experiments). (F) Distribution of the angle between actin and condensate movement vectors for pLAT → N-WASP$_{Basic}$ (gold), randomized pLAT → N-WASP$_{Basic}$ (purple), pLAT → N-WASP$_{WT}$ (black, same data as in *Figure 2*), pLAT → N-WASP$_{Neutral}$ (blue), and randomized pLAT → N-WASP$_{Neutral}$ (red). N = 15 fields of view from three independent experiments (5 FOV per experiment). P-values are for indicated distribution comparisons via Kolmogorov-Smirnov test.

DOI: https://doi.org/10.7554/eLife.42695.011

The following source data and figure supplements are available for figure 3:

**Source data 1.** Source data file for *Figure 3A*.
DOI: https://doi.org/10.7554/eLife.42695.028
**Source data 2.** Source data file for *Figure 3B*.
DOI: https://doi.org/10.7554/eLife.42695.029
**Source data 3.** Source data file for *Figure 3C*.
DOI: https://doi.org/10.7554/eLife.42695.030
**Figure supplement 1.** pLAT → Nck, pLAT → N-WASP, and pLAT → WASP condensates bind F-Actin.
DOI: https://doi.org/10.7554/eLife.42695.012
**Figure supplement 1—source data 1.** Source data file for *Figure 3—figure supplement 1*.
DOI: https://doi.org/10.7554/eLife.42695.013
**Figure supplement 2.** Quantitative analysis of actin filaments binding to LAT condensates.
DOI: https://doi.org/10.7554/eLife.42695.014
**Figure supplement 2—source data 1.** Source data file for *Figure 3—figure supplement 2*.
DOI: https://doi.org/10.7554/eLife.42695.015
**Figure supplement 2—source data 2.** Source data file for *Figure 3—figure supplement 2*.
DOI: https://doi.org/10.7554/eLife.42695.016
**Figure supplement 3.** Nck does not co-sediment with actin filaments.
DOI: https://doi.org/10.7554/eLife.42695.017
**Figure supplement 3—source data 1.** Source data file for *Figure 3—figure supplement 3*.
DOI: https://doi.org/10.7554/eLife.42695.018
**Figure supplement 4.** N-WASP binding partners do not increase N-WASP co-sedimentation with F-actin.
DOI: https://doi.org/10.7554/eLife.42695.019
**Figure supplement 4—source data 1.** Source data file for *Figure 3—figure supplement 4*.
DOI: https://doi.org/10.7554/eLife.42695.020
**Figure supplement 5.** The ability of His-tagged Nck variants to recruit actin filaments to an SLB depends on the number of basic regions (concentrated in L1 and the SH2 domain) vs. the number of acidic regions (concentrated in the second SH3 domain).
DOI: https://doi.org/10.7554/eLife.42695.021
**Figure supplement 5—source data 1.** Source data file for *Figure 3—figure supplement 5*.
DOI: https://doi.org/10.7554/eLife.42695.022
**Figure supplement 6.** His-tagged Nck variants containing mutations to L1 recruit actin to the bilayer in a density-dependent manner when L1 is WT or contains basic mutations.
DOI: https://doi.org/10.7554/eLife.42695.023
**Figure supplement 6—source data 1.** Source data file for *Figure 3—figure supplement 6*.
DOI: https://doi.org/10.7554/eLife.42695.024
**Figure supplement 7.** LAT condensates containing N-WASP fragments recruit actin to SLBs.
DOI: https://doi.org/10.7554/eLife.42695.025
**Figure supplement 8.** His-tagged N-WASP$_{Neutral}$, N-WASP$_{WT}$, or N-WASP$_{Basic}$ recruit actin filaments to SLBs in a density-dependent manner.
DOI: https://doi.org/10.7554/eLife.42695.026
**Figure supplement 8—source data 1.** Source data file for *Figure 3—figure supplement 8*.
DOI: https://doi.org/10.7554/eLife.42695.027

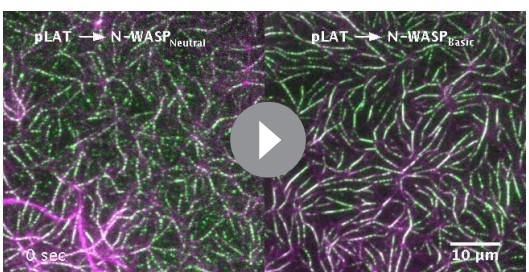

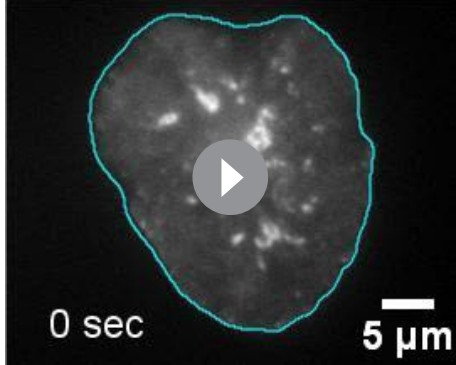

**Video 5.** Reconstitution of LAT condensate movement using different component mixtures containing N-WASP$_{Neutral}$ or N-WASP$_{Basic}$ in contracting actomyosin networks. TIRF microscopy revealed actin filament and LAT condensate movement on supported lipid bilayers in a contracting actomyosin network. His$_8$-pLAT-Alexa488 (green) was attached to Ni-functionalized SLBs at 500 molecules / μm$^2$. Actin filaments (magenta) were attached to the same bilayers via His$_{10}$-ezrin actin binding domains. Condensate component mixtures are indicated at the top of each movie panel. All reconstitutions were performed in 50 mM KCl. Condensates were formed by adding 125 nM Grb2, 62.5 nM Sos1, 62.5 nM pSLP-76, 125 nM Nck, and 125 nM N-WASP$_{Neutral}$ (pLAT → N-WASP$_{Neutral}$) or 125 nM Grb2, 62.5 nM Sos1, 62.5 nM pSLP-76, 125 nM Nck, and 125 nM N-WASP$_{Basic}$ (pLAT → N-WASP$_{Basic}$). After formation, both types of condensates bound to and wet actin filaments, although pLAT → N-WASP$_{Neutral}$ condensates did not wet filaments to the same degree as either pLAT → N-WASP$_{WT}$ condensates (compare with *Video 6*) or pLAT → N-WASP$_{Basic}$. Actin filament contraction was induced by adding 100 nM myosin II. pLAT→ N-WASP$_{Neutral}$ condensates did not efficiently move with contracting actin filaments (although they moved with actin filaments to a greater degree than pLAT → Sos1 condensates [compare with *Video 6*]) while pLAT → N-WASP$_{Basic}$ condensates efficiently moved with contracting actin filaments. Movie shows a 52 μm x 52 μm field of view for each movie panel. The movie is played at five fps with frame intervals of 5 s.
DOI: https://doi.org/10.7554/eLife.42695.031

**Video 6.** Single particle tracking of LAT condensates in activated Jurkat T cells. TIRF microscopy of an activated Jurkat T cell expressing LAT-mCherry on a SLB coated with ICAM-1 and OKT3. LAT condensates form at the synapse edge (cyan perimeter) and are tracked, using uTrack, as they move towards the cSMAC. Track colors: Cyan-track segment between adjacent frames; Red- gap closing between non-adjacent frames (maximum of 3 frames between end of a track segment and the beginning of another). Showing a maximum track tail length of 10 frames for visual clarity. The movie is played at five fps with frame intervals of 5 s.
DOI: https://doi.org/10.7554/eLife.42695.032

filaments proportionally to the degree of positive charge, and that these interactions are necessary for LAT condensates to faithfully move with actin.

## The composition of LAT condensates changes as they traverse the IS in cells

In Jurkat T cells activated by SLBs coated with ICAM-1 and OKT3, an antibody that recognizes the CD3ε subunit of the TCR, LAT condensates that form at the periphery of the IS move to its center over ~5 min as the cell-SLB contact matures. To investigate whether the composition-dependent interactions observed in our biochemical data might have consequences for the behavior of LAT condensates in cells, we examined the composition of LAT condensates as they moved in the plane of the plasma membrane during activation of live Jurkat T cells. We used Jurkat T cells because they retain many features of primary T cells relevant to movement and signaling from LAT condensates, but are easier to manipulate and analyze. In both cell types, LAT condensate formation is well-documented (*Balagopalan et al., 2013*; *Lin et al., 1999*; *Su et al., 2016*; *Yokosuka et al., 2005*), condensate movement across the IS is correlated with actin flow (*DeMond et al., 2008*; *Kaizuka et al., 2007*; *Murugesan et al., 2016*; *Yi et al., 2012*), and proximal biochemical signaling from LAT through SLP-76 is similar (*Bartelt et al., 2009*). However, the IS between Jurkat T cells and supported lipid bilayers is larger than that of primary T cells (*Murugesan et al., 2016*), and LAT condensates do not initiate actin polymerization to the same degree in Jurkat T cells as in primary T cells, which allows us to analyze the ability of condensates to couple to dynamic actin networks without accounting for their own self-generated polymerized actin (*Kumari et al., 2015*).

We co-expressed LAT-mCitrine with Grb2-mCherry or LAT-mCherry with Nck-sfGFP and LifeAct-BFP in Jurkat T cells. These cells bound to SLBs coated with mobile ICAM-1 and OKT3, producing an IS mimic with the SLB. In contrast to Jurkat T cells adhered to immobile substrates, which extend long lamellipodia (*Babich et al., 2012*), Jurkat T cells adhering to fluid SLBs here extend only short lamellipodia at the IS periphery. We used TIRF microscopy to capture images of activated cells every 5 s for up to 5 min. We then automatically detected and tracked LAT condensates from their formation at the periphery of the IS to their coalescence with the central supramolecular activation complex (cSMAC) at the synapse center (*Jaqaman et al., 2008*) (*Figure 4—figure supplement 1A*, *Figure 4—figure supplement 2*, *Video 6*), monitoring the fluorescence intensity of LAT and Grb2 or Nck. We tracked condensates based on the LAT channel, and then read out the Grb2 or Nck intensities at the condensate locations ('master/slave' channel analysis, as in *Loerke et al., 2011*). This scheme ensured accurate condensate detection and tracking because of the stronger signal in the LAT channel, especially when compared to the Nck channel, which tended to have high background due to soluble Nck molecules not associated with the membrane. To overcome the stochasticity and noise inherent to live-cell image data, we aligned tracks (in space or time, as described below) and averaged them to uncover underlying overall trends. For meaningful alignment and averaging, we filtered tracks by their duration, extent of directed movement, initial and final positions (to ensure that they traversed a sufficient radial distance across the IS), and initial Grb2 or Nck intensity to ensure that changes in intensity could be measured accurately (see Materials and methods for details).

Initially we aligned tracks according to radial position in the synapse, defining distance from the center of the synapse to the synapse edge as 'normalized radial position' (equal to zero at the synapse center and one at the synapse edge; see Materials and methods). This analysis revealed that Grb2 colocalized with LAT condensates at the edge of the synapse (*Figure 4A*, *Video 7*) and its fluorescence intensity in the condensates was maintained throughout the trajectory to the center of the synapse (*Figure 4B,C*). In contrast, while Nck also colocalized with LAT condensates at the edge of the synapse (*Figure 4D*, *Video 8*), its fluorescence diminished relative to LAT during the trajectory. The rate of change was largest in the ~0.8–0.5 interval of normalized radial positions. The decrease in the Nck:LAT ratio within condensates compared to the peak value was statistically significant starting at the 0.7–0.6 normalized radial position interval, based on a Rank-Sum test of the medians (*Figure 4C,E*). The change in composition slowed down to an almost negligible rate after the 0.6–0.5 normalized radial position interval, that is in the central half of the IS (*Figure 4C*). The emergence of this pattern from averaging 125 spatially aligned tracks from 25 cells suggests that spatial position is a key determinant of Nck residence in LAT condensates. In contrast, when the condensates were aligned according to the time of their appearance (*Figure 4—figure supplement 1B*), the composition measurements per time point were less well defined (as reflected by larger 95% confidence intervals around the median), the pattern over time was noisier, and the median Nck intensity more or less steadily decreased over the course of the trajectory. The decrease in Nck intensity is not the result of photobleaching (*Figure 4—figure supplement 3*). This suggests that in this experimental setting, position plays a more instructive role than time in determining the residence of Nck in condensates (and presumably the residence of WASP, which is recruited to LAT condensates via Nck). However, since time and space are coupled, our data do not rule out a role for time in this process, as has previously been observed in experimental conditions where LAT condensates were immobilized (*Barda-Saad et al., 2005*; *Yi et al., 2019*).

## Nck dissipation spatially parallels the changes in actin architecture from dendritic network to concentric arcs

Translocation of LAT condensates from the synapse periphery to the center of the IS is driven by motion of the actin cytoskeleton (*DeMond et al., 2008*; *Kaizuka et al., 2007*; *Mossman et al., 2005*; *Yu et al., 2010*). Recent work has shown that two actin networks are generated at the IS in activated T cells (*Murugesan et al., 2016*; *Yi et al., 2012*). The outer ~1/3 of the synapse is enriched in a dendritic actin meshwork generated by the Arp2/3 complex, where the filaments are on average directed radially, perpendicular to the synapse edge. Progressing toward the cSMAC this meshwork is replaced by formin-generated concentric actin arcs that are directed parallel to the synapse edge (*Figure 4—figure supplement 4*) (*Hammer and Burkhardt, 2013*; *Yi et al., 2012*). The arcs dominate in the central ~1/2 of the IS. Both filament networks move through the action of myosin motors as the cell-cell conjugate matures; however, the nature of this movement is different in the two

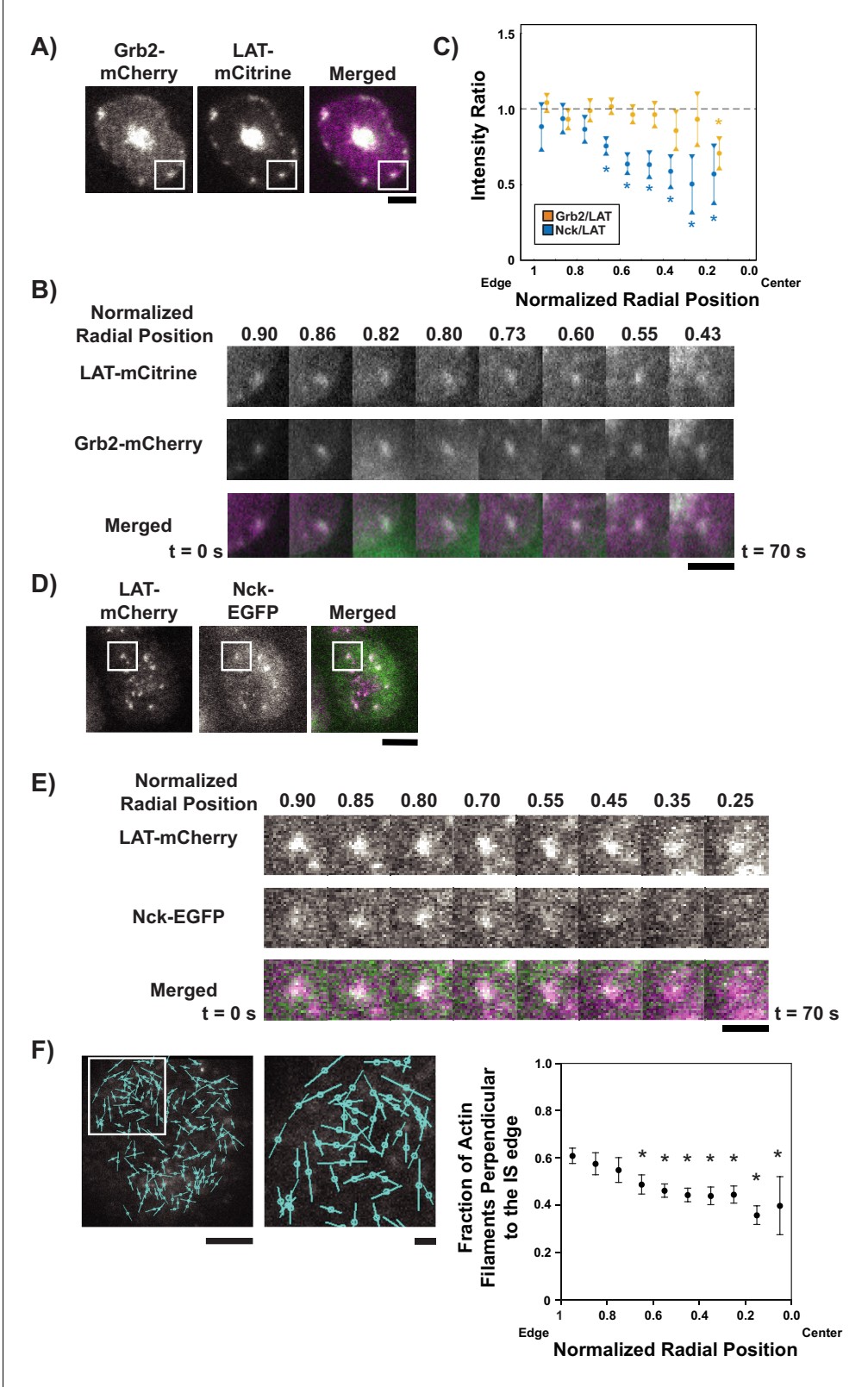

**Figure 4.** LAT condensates change composition as they move across the IS. (**A**) TIRF microscopy image of Jurkat T cell expressing Grb2-mCherry (magenta in merge) and LAT-mCitrine (green in merge) activated on an SLB coated with OKT3 and ICAM-1. Scale Bar = 5 µm. (**B**) Magnification of boxed region in (**A**), with normalized radial position (one at synapse edge, 0 at cSMAC center) indicated above images and time indicated below. Scale bar = 2 µm. *Figure 4 continued on next page*

*Figure 4 continued*

(C) Quantification of fluorescence intensity ratios of Grb2/LAT (gold) and Nck/LAT (blue) in LAT condensates in Jurkat T cells at different normalized radial positions. Measurements were made at identical relative locations but data points are slightly offset in the graph for visual clarity. Plot displays median and its 95% confidence interval from N = 125 condensates from 25 cells expressing Nck-sfGFP and LAT-mCherry from five independent experiments and 82 condensates from 11 cells expressing Grb2-mCherry and LAT-mCitrine from five independent experiments. Only tracks in which the mean Grb2 or Nck intensity was greater than one standard deviation above background during the first three measurements were used for this analysis. Asterisks indicate data points whose values differ significantly from the reference data point (radial position = 1.0–0.9 for Grb2 and radial position = 0.9–0.8 for Nck) as determined using a Wilcoxon rank-sum test with Bonferroni correction. The Bonferroni correction was used to achieve a total type-I error of 0.05. For eight comparisons, this leads to a significance threshold per pair = 0.006. (D) TIRF microscopy image of Jurkat T cell expressing Nck-sfGFP (green in merge) and LAT-mCherry (magenta in merge) activated on an SLB by OKT3 and ICAM-1. Scale Bar = 5 µm. (E) Magnification of boxed region in (D). Scale bars and details as in (B). (F) Fluorescence polarization microscopy image of a Jurkat T cell sparsely labeled with SiR-Actin and activated on an SLB coated with OKT3 and ICAM-1 (left, scale bar = 10 µm), magnification of boxed region in image (middle, scale bar = 2 µm), and fraction of filaments perpendicular (+ / - 45°) to the synapse edge (right). Cyan lines in left and center images indicate the orientation of the fluorescence dipole of the SiR fluorophore which in turn is oriented orthogonal to the underlying actin filament (see *Figure 4—figure supplement 5*). Shown are the mean ±s.d. from N = 13,052 particles from 6 cells. Asterisks indicate data points whose values differ significantly from the reference data point (radial position = 1.0–0.9) as determined using a t-test with Bonferroni correction to achieve a total type-I error of 0.05.
DOI: https://doi.org/10.7554/eLife.42695.033

The following source data and figure supplements are available for figure 4:

**Source data 1.** Source data file for *Figure 4F*.
DOI: https://doi.org/10.7554/eLife.42695.040
**Figure supplement 1.** Nck dissipation from condensates is spatially regulated within the first 5 min of synapse formation.
DOI: https://doi.org/10.7554/eLife.42695.034
**Figure supplement 2.** Image analysis for live cell data.
DOI: https://doi.org/10.7554/eLife.42695.035
**Figure supplement 3.** Photobleaching analysis of live-cell data.
DOI: https://doi.org/10.7554/eLife.42695.036
**Figure supplement 4.** Two actin networks at the IS, with different architecture and movement.
DOI: https://doi.org/10.7554/eLife.42695.037
**Figure supplement 5.** SiR-Actin polarity is perpendicular to actin filaments.
DOI: https://doi.org/10.7554/eLife.42695.038
**Figure supplement 6.** The two-dimensional extension of the dense actin network at the IS in Jurkat T cells activated on SLBs is independent of the three dimensional shape of the activated Jurkat T cell.
DOI: https://doi.org/10.7554/eLife.42695.039

cases. The outer dendritic network moves in a direction perpendicular to the edge of the synapse in a process termed retrograde flow (*DeMond et al., 2008*; *Kaizuka et al., 2007*; *Mossman et al., 2005*; *Yu et al., 2010*), analogous to actin flow observed at the leading edge of migrating cells (*Ponti et al., 2005*; *Ponti et al., 2004*). In contrast, the inner concentric arcs sweep toward the center of the synapse in a telescoping manner and appear to have components of motion both perpendicular and parallel to the synapse edge (*Murugesan et al., 2016*).

The spatial pattern of Nck dissipation from LAT condensates is similar to the reported actin transition from dendritic architecture to arc architecture (*Hammer and Burkhardt, 2013*; *Murugesan et al., 2016*; *Yi et al., 2012*). To corroborate this in our cells, we incubated Jurkat T cells with the dye SiR-Actin, which binds to actin filaments with a defined orientation (perpendicular to the filament orientation; *Figure 4—figure supplement 5*) (*Nordenfelt et al., 2017*). This dye enabled us to use instantaneous polarization TIRF microscopy (*Mehta et al., 2016*) to evaluate the orientation of actin filaments at the IS. We found that SiR-actin at concentrations higher than 50 nM blocked actin flow at the IS, but speckle labeling with 10 nM SiR-actin allowed for actin flow from the edge to the center (data not shown). Since the actin networks were stained as sparsely distributed speckles of SiR-actin, our analysis procedure does not involve (or require) identification of individual filaments in the images to characterize their orientations. Rather, by observing the

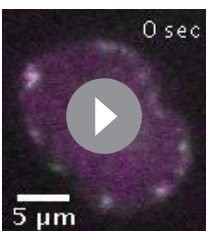

**Video 7.** Grb2 is maintained as LAT condensates move across the IS in activated Jurkat T cells. TIRF microscopy of an activated Jurkat T cell expressing Grb2-mCherry (magenta) and LAT-mCitrine (green) on a SLB coated with ICAM-1 and OKT3 revealed that Grb2 co-localizes with LAT condensates as they move from the edge of the synapse to the cSMAC. Movie shows a 22 μm x 22 μm field of view. The movie is played at seven fps with frame intervals of 5 s.
DOI: https://doi.org/10.7554/eLife.42695.041

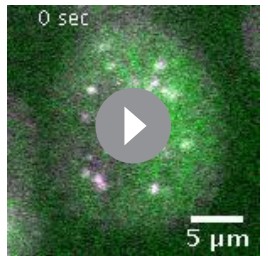

**Video 8.** Nck dissipates from LAT condensates as they move across the IS in activated Jurkat T cells. TIRF microscopy of an activated Jurkat T cell expressing LAT-mCherry (magenta) and Nck-sfEGFP (green) on a SLB coated with ICAM-1 and OKT3 revealed that Nck dissipates from LAT condensates as they move from the edge of the synapse to the cSMAC. Movie shows a 24 μm x 24 μm field of view. The movie is played at five fps with frame intervals of 5 s.
DOI: https://doi.org/10.7554/eLife.42695.042

fluorescence polarization orientation of single speckles randomly bound to actin filaments, we can build a spatial map of filament orientations across the IS. We found that actin filaments in the outer 30% of the synapse were generally oriented perpendicular to the synapse edge, while those closer to the center of the synapse were parallel to the synapse edge (*Figure 4F*), in good agreement with earlier super-resolution work (*Murugesan et al., 2016*). The rate of change of orientation was largest in the ~0.8–0.6 interval of normalized radial positions. T-test statistical analysis, in which actin filament orientation at each spatial point was compared with filament orientation at the edge of the IS, demonstrated a significant change in actin filament orientation relative to the edge starting at a normalized radial position of 0.7–0.6 and was maintained to the center of the IS. This spatial pattern of actin filament orientation correlates well with the spatial pattern of Nck dissipation from LAT condensates (compare *Figure 4C and F*). The location of the changes in actin orientation is unrelated to the overall three-dimensional geometry of the cell, indicating that actin architecture at the IS, as observed via TIRF microscopy, is not determined by the position of the cell body above the membrane surface (*Figure 4—figure supplement 6*).

These data show that the changes in the actin network at the IS, from a peripheral dendritic organization to a more central collection of circular arcs, are spatially correlated with the changes in composition of the LAT condensates, from high levels of Nck enrichment at the periphery to low levels nearer the center.

## Constitutive engagement of LAT condensates with actin leads to their aberrant movement across the IS

Our combined biochemical and cellular data thus far indicate that Nck and WASP mediate LAT condensate engagement with actin, and that LAT condensates lose these proteins as they move from the dendritic actin meshwork in the outer part of the synapse to the contractile arcs closer to the synapse center. The biochemical data suggest that this change in composition should allow LAT condensates to interact differently with the two actin networks, with stronger adhesion to the dendritic meshwork and weaker adhesion to the contractile arcs. We hypothesized that this change in interaction might be necessary for the proper radial movement of LAT condensates at the IS, given the different orientation (perpendicular vs. parallel to the synapse edge) and movement (retrograde flow vs. telescoping motion) of filaments in the two actin networks. To test this hypothesis, we altered the adhesion of LAT condensates to the actin filament network by fusing Grb2, which remains in the condensates throughout their trajectories (*Figure 4A–C*), with the doubled basic region of N-WASP (Grb2$_{Basic}$).

In biochemical assays, this fusion protein generated LAT condensates that bound actin filaments in the absence of Nck or WASP. The pLAT-Grb2$_{basic}$ complex recruited actin filaments to SLBs, while

pLAT alone or the pLAT-Grb2 complex did not (*Figure 5A*). In actomyosin contraction assays, condensates of pLAT/Grb2$_{Basic}$/Sos1 initially wet filaments and then localized to actin asters after myosin II-induced contraction to a greater degree than pLAT/Grb2/Sos1, although to a lesser degree than pLAT → N-WASP (*Figure 5B*, *Video 9* vs. *Video 4*). Similarly, during actomyosin network contraction, the movement of pLAT/Grb2$_{Basic}$/Sos1 condensates was correlated more strongly with actin movement than condensates containing Grb2 (*Figure 5—figure supplement 1*), but less strongly than pLAT → N-WASP condensates (*Figure 2C,D*). Together, these data demonstrate that the double basic motif of N-WASP, when added to Grb2, can act as a molecular clutch, coupling LAT condensates to actin in vitro.

We next asked whether expression of Grb2$_{Basic}$ in Jurkat T cells would perturb the radial movement of LAT condensates due to their constitutively engaged clutch, including in the medial region of the IS where they encounter actin arcs. We quantified the deviation from a straight path of condensate trajectories between the start of persistent inward radial movement and coalescence with the cSMAC (see Supplemental Methods for details). For these experiments, cells expressing Grb2$_{Basic}$-mCherry were activated on SLBs as above. Similar to cells expressing Grb2-mCherry, LAT condensates that formed at the periphery of the IS retained Grb2$_{Basic}$-mCherry throughout their trajectories to the cSMAC (*Figure 5C,D*). However, statistical analysis of these trajectories demonstrated a small, but significant, deviation from a straight path when compared with condensates in cells expressing Grb2-mCherry (*Figure 5E,F*, *Video 10* vs. *Video 7*). This behavior is consistent with abnormally high adhesion of condensates containing Grb2$_{Basic}$ to actin filaments, even after Nck has presumably dissipated, leading to trajectories that reflected more the telescoping, circular component of the contractile actin arc motion.

## Formin activity is necessary for LAT condensate composition change

Finally, we asked whether the transition from the dendritic actin architecture to the contractile arcs

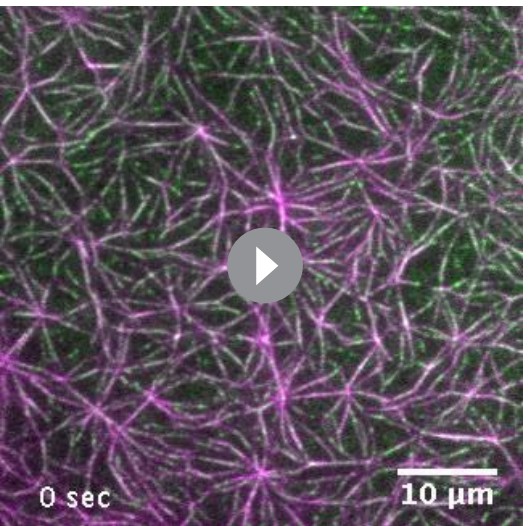

**Video 9.** Reconstitution of movement in LAT condensates containing Grb2$_{Basic}$ in contracting actomyosin networks. TIRF microscopy revealed actin filament and LAT condensate movement on supported lipid bilayers in a contracting actomyosin network. His$_8$-pLAT-Alexa488 (green) was attached to Ni-functionalized SLBs at 500 molecules / µm$^2$. Actin filaments (magenta) were attached to the same bilayers via His$_{10}$-ezrin actin binding domains. Reconstitution was performed in 50 mM KCl. Condensates were formed by adding 125 nM Grb2$_{Basic}$ and 125 nM Sos1. After formation, condensates bound to and wet actin filaments. Actin filament contraction was induced by adding 100 nM myosin II. LAT condensates composed of Grb2$_{Basic}$ moved with contracting actin filaments, although not to the same degree as pLAT → N-WASP$_{WT}$ condensates (Compare with *Video 6*). Movie shows a 52 µm x 52 µm field of view. The movie is played at five fps with frame intervals of 5 s.
DOI: https://doi.org/10.7554/eLife.42695.045

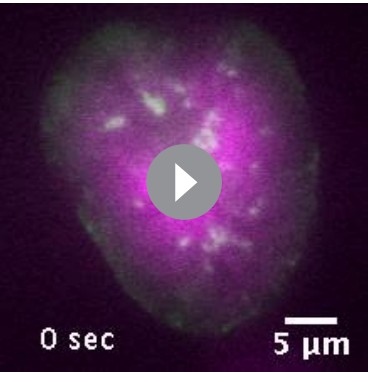

**Video 10.** LAT condensates containing Grb2$_{Basic}$ deviate tend to deviate from a radial trajectory across the IS. TIRF microscopy of an activated Jurkat T cell expressing Grb2$_{Basic}$-mCherry and LAT-mCitrine on a SLB coated with ICAM-1 and OKT3 revealed that condensates containing Grb2$_{Basic}$-mCherry tend to deviate from a radial trajectory as they move across the IS (Compare with *Video 1*). Movie shows a 35 µm x 35 µm field of view. The movie is played at six fps with frame intervals of 5 s.
DOI: https://doi.org/10.7554/eLife.42695.046

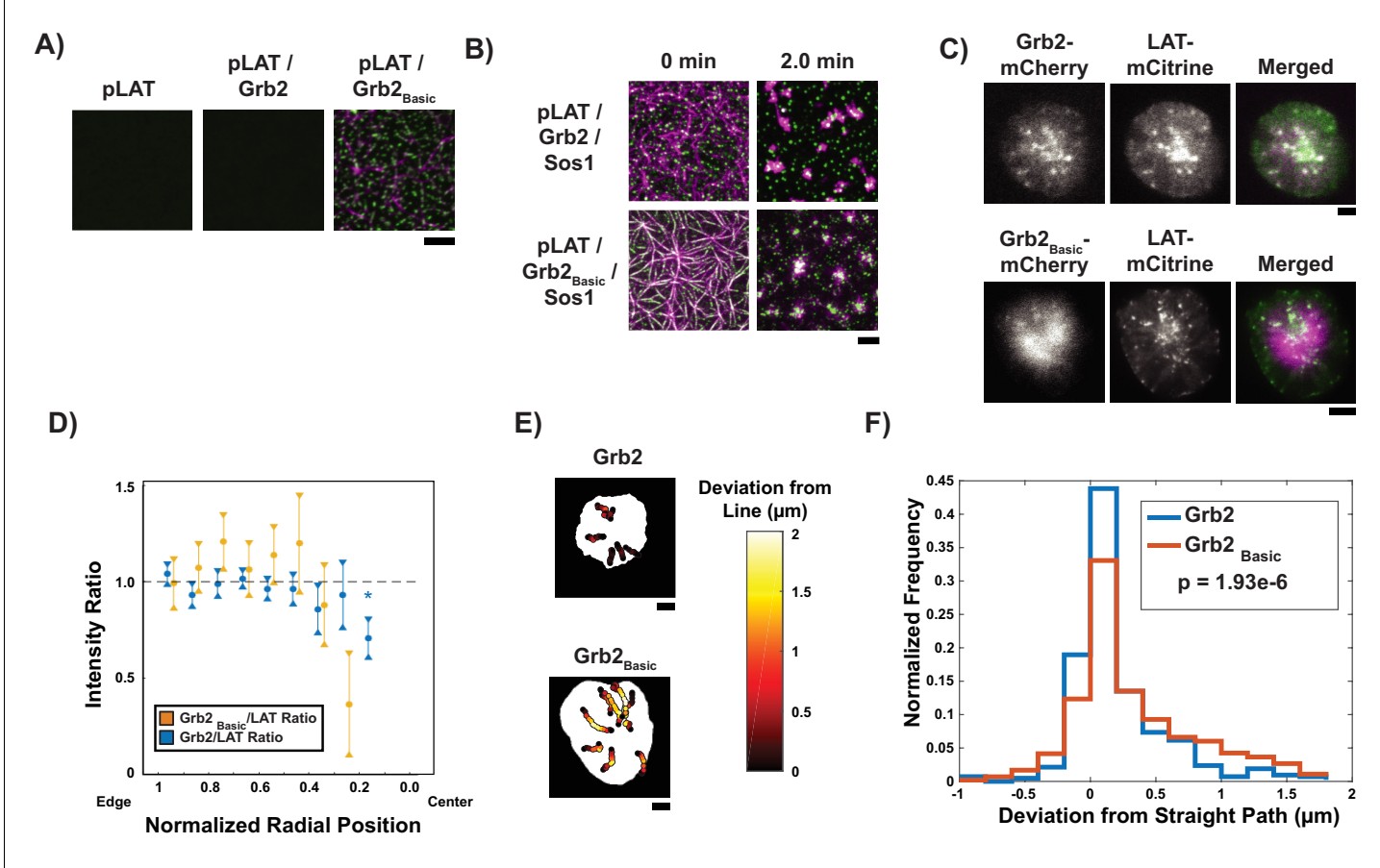

**Figure 5.** Grb2 fused to a basic molecular clutch can couple LAT condensates to actin. (A) TIRF microscopy images of rhodamine-actin recruited to SLBs by His-tagged pLAT or condensates of pLAT → Grb2 or pLAT → Grb2$_{Basic}$. Scale bar = 5 µm. (B) TIRF microscopy images of pLAT → Sos1 condensates containing Grb2$_{WT}$ (top row; data from *Figure 2*) or Grb2$_{Basic}$ (bottom row) formed in an actin network before (t = 0 min) and after (t = 2 min) addition of myosin II. Actin shown in magenta and LAT condensates in green. Scale bar = 5 µm. (C) TIRF microscopy image of Jurkat T cell expressing Grb2-mCherry (top, magenta in merge) or Grb2$_{Basic}$-mCherry (bottom, magenta in merge) and LAT-mCitrine (green in merge) activated on an SLB coated with OKT3 and ICAM-1. Scale Bar = 5 µm. (D) Fluorescence intensity ratios of Grb2/LAT (blue, data from *Figure 4C*) or Grb2$_{Basic}$ / LAT in condensates (gold) at different normalized radial positions. Measurements were made at identical relative locations but data are slightly offset in the graph for visual clarity. Plot displays median and notches from boxplot for 95% confidence interval from N = 44 condensates from 12 cells expressing Grb2$_{Basic}$-mCherry and LAT-mCitrine from seven independent experiments and 82 condensates from 11 cells expressing Grb2-mCherry and LAT-mCitrine from five independent experiments (same cells as in *Figure 4C*). Only tracks in which the mean Grb2 or Grb2$_{Basic}$ intensity was greater than one standard deviation above background during the first three measurements were used to generate this plot. Asterisk indicates data point whose value differs significantly from the reference data point (radial position = 1.0–0.9) as determined using a Wilcoxon rank-sum test with Bonferroni correction. The Bonferroni correction was used to achieve a total type-I error of 0.05. For eight comparisons, this leads to a significance threshold per pair = 0.006. (E) Trajectories of LAT condensates in Jurkat T cells expressing Grb2-mCherry (top) or Grb2$_{Basic}$-mCherry (bottom) recorded over 2 to 5 min of imaging. Trajectories are color-coded as indicated in the legend at right according to deviation from a straight line between the estimated starting point of actin engagement and just before entering the cSMAC (see Materials and methods). Scale bar = 5 µm. (F) Distribution of deviations from a straight line for condensates in Jurkat T cells expressing Grb2-mCherry (red) or Grb2$_{Basic}$-mCherry (blue). N = 44 condensates from 12 cells expressing Grb2$_{Basic}$-mCherry and LAT-mCitrine from seven independent experiments and 82 condensates from 11 cells expressing Grb2-mCherry and LAT-mCitrine from independent experiments (same cells as in *Figure 4C*). P-value is for comparing the two distributions via a Kolmogorov-Smirnov test. Only tracks in which the mean Grb2 or Grb2$_{Basic}$ intensity was greater than one standard deviation above background during the first three measurements were used to generate this plot.

DOI: https://doi.org/10.7554/eLife.42695.043

The following figure supplement is available for figure 5:

**Figure supplement 1.** Adding a doubled N-WASP basic domain to Grb2 (Grb2$_{Basic}$) enables pLAT / Grb2$_{Basic}$/Sos1 condensates to move with actin to a better degree than condensates formed with WT Grb2.

DOI: https://doi.org/10.7554/eLife.42695.044

might play a role in changing the composition of LAT condensates. Previous data showed that the contractile arcs are generated by the formin mDia1 and could be eliminated by the formin inhibitor, SMIFH2 (*Murugesan et al., 2016*). We found that in contrast to control cells treated with DMSO, where Nck dissipated normally from LAT condensates (*Figure 6A and B*, *Figure 6—figure supplement 1*, *Video 11*), cells treated with SMIFH2 for five minutes prior to imaging displayed LAT condensates with virtually constant Nck intensities throughout their trajectories from the periphery to the cSMAC (*Figure 6A and C*, *Figure 6—figure supplement 1*, *Video 12*). Thus, the activity of formin proteins, and/or perhaps the actin arcs that they generate, act to alter the composition of LAT condensates, likely altering their downstream signaling activities in the central region of the IS. We note that the SMIFH2 data further support the notion that in unperturbed cells, space, rather than time, is the key determinant of Nck residence in condensates (assuming that formins do not also create a temporal signal). Our combined data suggest that the two actin networks in activated Jurkat T

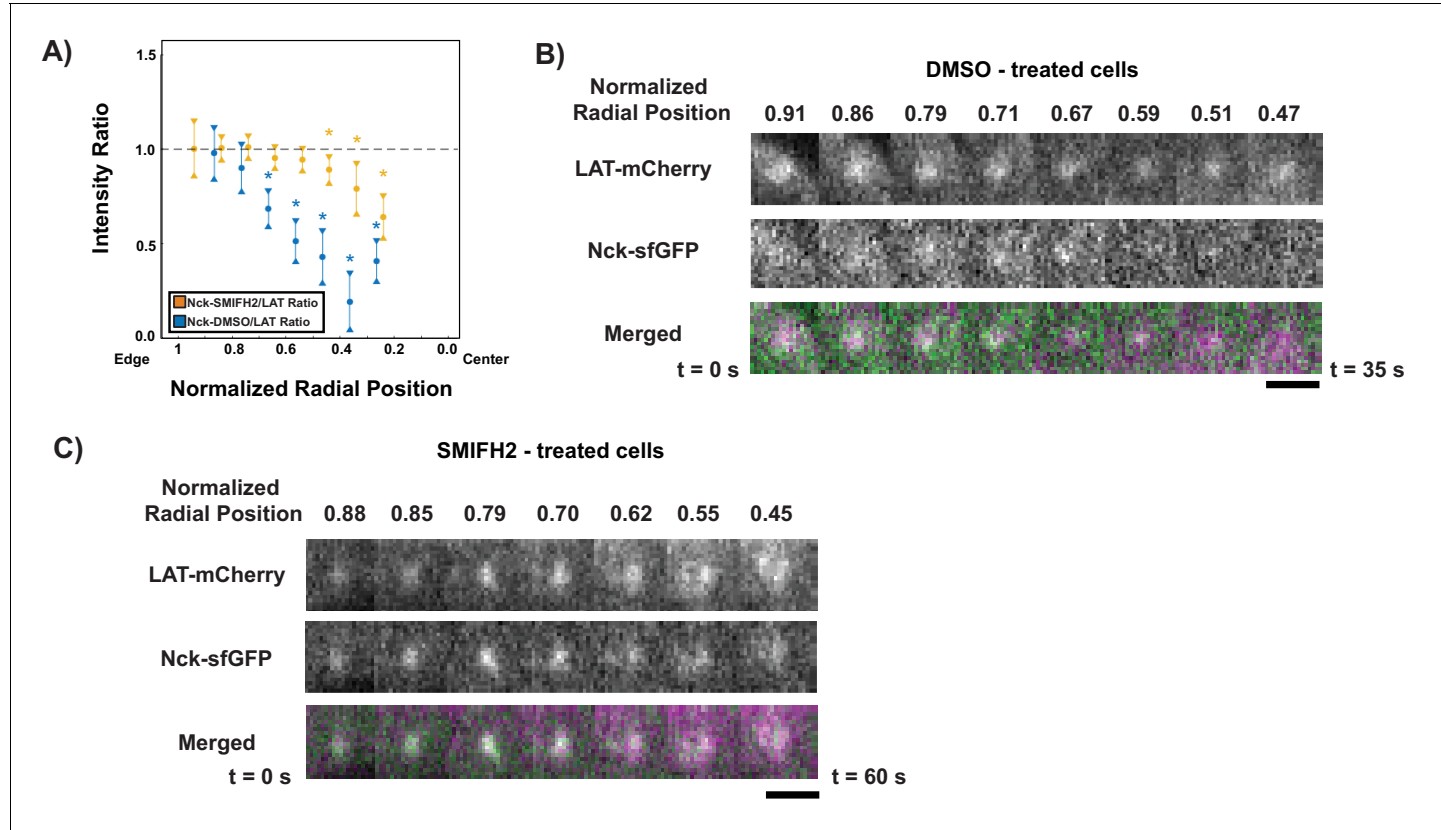

**Figure 6.** Formin activity is necessary for Nck dissipation from LAT condensates. (A) Fluorescence intensity ratios of Nck/LAT in condensates in Jurkat T cells treated with DMSO (blue) or the formin inhibitor, SMIFH2, (gold) at different normalized radial positions. Measurements were made at identical relative locations but data are slightly offset in the graph for visual clarity. Plot displays median and notches from boxplot for 95% confidence interval from N = 43 condensates from 11 DMSO-treated cells from five individual experiments and 102 condensates from 14 SMIFH2-treated cells from five individual experiments. Only tracks in which the mean Nck intensity was greater than one standard deviation above background during the first three measurements were used to generate this plot. The first DMSO data point (radial position = 1.0–0.9) does not appear in the plot because the number of detected condensates was too small (<10) to generate a statistically meaningful measurement. Asterisks indicate data points whose values differ significantly from the reference data point (radial position = 0.9–0.8) as determined using a Wilcoxon rank-sum test with Bonferroni correction. The Bonferroni correction was used to achieve a total type-I error of 0.05. For eight comparisons, this leads to a significance threshold per pair = 0.006. (B, C) Magnification of boxed regions from *Figure 6—figure supplement 1* of condensates containing LAT-mCherry (magenta in merge) and Nck-sfGFP (green in merge) during their trajectories across the IS in a cell treated with DMSO (B) or SMIFH2 (C). Normalized radial position indicated above image panels and time below panels. Scale bar = 2 µm.

DOI: https://doi.org/10.7554/eLife.42695.047

The following figure supplement is available for figure 6:

**Figure supplement 1.** Example images showing DMSO- or SMIFH2-treated Jurkat T cells activated on SLBs.
DOI: https://doi.org/10.7554/eLife.42695.048

cells not only spatially organize the immunological synapse by moving LAT condensates, but may also contribute to creation of specific signaling zones.

## Discussion

Compositional changes of biomolecular condensates in response to signals have been well documented (*Chen et al., 2008*; *Dellaire et al., 2006*; *Markmiller et al., 2018*; *Salsman et al., 2017*; *Youn et al., 2018*). However, the functional consequences of these compositional changes have generally not been elucidated. Here, we show that compositional changes alter the interactions of LAT condensates on membranes with active actomyosin networks. Our in vitro data demonstrate that Nck, WASP, and N-WASP act as a clutch that mediates strong binding of LAT condensate to actin filaments, thus coupling movement of the structures to movement of actomyosin filaments. LAT condensates lacking Nck and WASP or N-WASP can still be propelled by other, possibly steric, interactions with moving actomyosin filaments, but less efficiently than when the clutch is present. We also show that the composition of LAT condensates in activated Jurkat T cells changes as they move radially from the edge to the center of the IS. This change spatially parallels the transition from the peripheral dendritic actin network to the circular actin arcs, and inhibition of the formin mDia1 prevents loss of Nck from LAT condensates. Mutations that add a basic sequence to Grb2, thus constitutively engaging LAT condensates with actin filaments, cause aberrant movement of the condensates with a higher tendency to deviate from a straight line.

These data suggest a potential model for movement of LAT condensates across the IS by navigating the distinct cortical actin networks. Condensates form in the outer region of the IS with the full complement of signaling molecules, including Grb2, Sos1, SLP-76, Nck, and WASP. Within this collection, the basic regions on Nck and WASP could act as a molecular clutch, enabling the condensates to adhere tightly to the outer dendritic actin network, and to travel radially with the network as it moves by retrograde flow. In the transition to the formin-generated actin arcs, a formin-dependent signal causes loss of Nck, and likely WASP, which is present in condensates largely through its interactions with Nck (*Cannon et al., 2001*) (although interactions between TCR, DOCK-8, WIP, and WASP could also contribute [*Janssen et al., 2016*]). We do not yet know the nature of this signal, but one possibility would be dephosphorylation of SLP-76, as Nck is known to join condensates primarily through binding SLP-76 phosphotyrosines (*Barda-Saad et al., 2010*; *Pauker et al., 2012*). Other mechanisms involving weakening of different interactions, appearance of competing Nck binding partners, or mechanical disruptions are also possible. Our biochemical data suggest that loss of Nck/WASP should decrease adhesion of the LAT condensates to actin. In this state, the condensates should still be movable by actin, based on our in vitro data on LAT → Sos1 condensates, but likely through more transient, weaker contacts. This is consistent with previous observations that in the actin arcs region of the IS, LAT condensates are repeatedly hit by arcs that move them briefly but then release (*Murugesan et al., 2016*). The circular movement of the telescoping actin arcs is randomly directed clockwise and counterclockwise, and repeated hits by arcs moving oppositely should produce no net circular motion on the condensates. But the radial component is consistently directed toward the center of the IS, and thus repeated hits could constructively produce a net movement in a radial direction. Thus, this model predicts that LAT condensates would continue to move linearly toward the center of the IS in the arc region. If the condensates were to adhere tightly to actin in the arcs region (i.e. if they contained Nck and WASP there), they would no longer undergo repeated hits by arcs moving in opposite directions, and the circumferential force would not average to zero. In the simplest case, condensates would attach to the first arc they encounter and move circumferentially with it. Such effects could account for the aberrant movement of LAT condensates containing the Grb2 mutant artificially equipped with a basic clutch, which should produce inappropriately strong adhesion to the telescoping actin arcs. Future cellular studies will directly test our proposed molecular clutch, in part by analysis of condensates lacking basic elements of Nck and WASP, and their movement in the periphery of the IS.

While we have examined Jurkat T cells here, existing data suggest that the behaviors we have described and the model we have proposed are likely relevant to primary T cells as well. The IS formed by both Jurkat T cells and primary T cells is composed of a dendritically-branched actin network at the edge of the IS, followed immediately by a concentric actomyosin cable network near the center of the IS (*Murugesan et al., 2016*). Thus, condensates must be moved across two distinct

actin networks in Jurkat and primary T cells. In primary T cells, condensate-associated actin polymerization localizes mostly (although not entirely) in the outer region of the IS and dissipates in the region adjacent to the branched actin network, where ICAM-1 localizes (*Kumari et al., 2015*), which would be consistent with loss of Nck and WASP toward the center of the IS. Similarly, PLC-γ1 activation by WASP-promoted actin polymerization appears to localize mostly in the dSMAC where we observe strong Nck co-localization with LAT (*Kumari et al., 2015*). One difference between the two cell types is that WASP-promoted actin polymerization is much weaker in Jurkat T cells than in primary T cells (*Kumari et al., 2015*). In primary cells, this actin assembly may also play a role in the movement of condensates from the cell edge to the center, in addition to myosin-driven movement of the cortical actin. Future work addressing the modes of movement, and the precise signals that dictate compositional change, will elucidate the mechanisms by which LAT condensates move across the IS in primary T cells.

LAT condensates represent one particular type of biomolecular condensate. It is generally thought that the functions of condensates are intimately connected to their compositions, and that changes in composition could cause changes in function (*Banani et al., 2016*). Our data here demonstrate that when Nck and N-WASP are arrayed on membranes they can bind actin filaments efficiently, even though both bind filament sides only weakly in solution. This adhesion enables condensates containing the proteins to be moved over long distances in response to actomyosin contraction. Adhesion is lost when Nck and WASP or N-WASP depart. Thus, the composition of LAT condensates plays an important role in their coupling to actin and their mode of movement at the IS. These behaviors of the LAT system are produced by generalizable features of membrane-associated condensates - their high density and composition based on regulatable interactions. Analogous behaviors are likely to be widely observed as the biochemical and cellular activities of other condensates are explored.

## Materials and methods

### Protein reagents

Human Ezrin (aa 477–586) with an N-terminal $His_{10}$-KCK tag, human Grb2 (aa 1–217), human Grb2-Double Basic (aa 1–217 of Grb2 fused with the human N-WASP basic region (x2) KEKKKGKAKKK RLTKGKEKKKGKAKKKRITK), human LAT (aa 48–233) with an N-terminal $His_8$ tag, human Nck1 (aa 1–377), human Nck (FL) (aa 1–377) with an N-terminal $His_8$-C(GGS)$_4$ tag, human Nck (FLΔL1) with an N-terminal $His_8$-C(GGS)$_4$ tag, human Nck (L1-S2-L2-S3-L3-SH2) with an N-terminal $His_8$-C(GGS)$_4$ tag, human Nck (L1(K to E)-S2-L2-S3-L3-SH2) with an N-terminal $His_8$-C(GGS)$_4$ tag, human Nck (L1(HM)-S2-L2-S3-L3-SH2) with an N-terminal $His_8$-C(GGS)$_4$ tag, human Nck (S1-L1-S3-L3-SH2) with an N-terminal $His_8$-C(GGS)$_4$ tag, human Nck (S1-L1-S2-L2-S3) with an N-terminal $His_8$-C(GGS)$_4$ tag, human Nck (S2-L2-S3-L3-SH2) with an N-terminal $His_8$-C(GGS)$_4$ tag, human Nck (S3-L3-SH2) with an N-terminal $His_8$-C(GGS)$_4$ tag, human Nck (L3-SH2) with an N-terminal $His_8$-C(GGS)$_4$ tag, human Nck (S1) with an N-terminal $His_8$-C(GGS)$_4$ tag, human Nck (S2) with an N-terminal $His_8$-C(GGS)$_4$ tag, human Nck (S3) with an N-terminal $His_8$-C(GGS)$_4$ tag, human Nck (FL(L1 KtoE)) with an N-terminal $His_8$-C(GGS)$_4$ tag, human Nck (FL(L1 basic)) with an N-terminal $His_8$-C(GGS)$_4$ tag, human Nck (FL(L1 (GGSA)$_{10}$)) with an N-terminal $His_8$-C(GGS)$_4$ tag, human SLP-76 (aa 101–420), human Sos1 (aa 1117–1319), human N-WASP (aa 451–485 of human WIP fused to aa 26–505 of human N-WASP), human N-WASP (aa 451–485 of human WIP fused to aa 26–505 of human N-WASP) with an N-terminal $His_6$ tag, human N-WASP (aa 451–485 of human WIP fused to aa 26–185 and 201–505 of human N-WASP) with an N-terminal $His_6$ tag and the basic region (KEKKKGKAKKKRLTK) doubled to (KEKKKGKAKKKRLTKGKEKKKGKAKKKRITK), human N-WASP (aa 451–485 of WIP fused to aa 26–185 and 201–505) with an N-terminal $His_6$ tag and the basic region (KEKKKGKAKKKRLTK) replaced with a (GGS)$_5$ linker, human N-WASP (aa 451–485 of human WIP fused to aa 26–185 and 201–505 of human N-WASP) with the basic region (KEKKKGKAKKKRLTK) doubled to (KEKKKGKAKKK RLTKGKEKKKGKAKKKRITK), human N-WASP (aa 451–485 of WIP fused to aa 26–185 and 201–505) with the basic region (KEKKKGKAKKKRLTK) replaced with a (GGS)$_5$ linker, and human WASP (aa 451–485 of human WIP fused to aa 39–502 of human WASP) were expressed and purified from bacteria. Actin was purified from rabbit skeletal muscle. Myosin II was purified from chicken skeletal

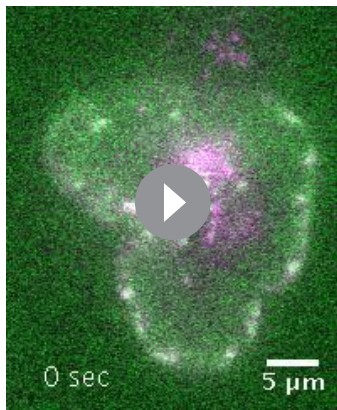

**Video 11.** Nck dissipates from LAT condensates as they move across the IS in activated Jurkat T cells treated with DMSO. TIRF microscopy of an activated Jurkat T cell expressing LAT-mCherry (magenta) and Nck-sfGFP (green) on a SLB coated with ICAM-1 and OKT3 revealed that Nck dissipates from LAT condensates as they move from the edge of the synapse to the cSMAC following treatment with DMSO for 5 min prior to activation. Movie shows a 31 µm x 31 µm field of view. The movie is played at six fps with frame intervals of 5 s.
DOI: https://doi.org/10.7554/eLife.42695.049

muscle. Rhodamine-labeled actin was purchased from Cytoskeleton, Inc. Details of constructs used in this study are listed in the *Table 1*.

## Protein purification and modification

### Ezrin purification

BL21(DE3) cells containing MBP-His$_{10}$-Ezrin Actin Binding Domain (ABD) were collected by centrifugation and lysed by sonication in 50 mM NaH$_2$PO$_4$, 10 mM imidazole (pH 7.5), 150 mM NaCl, and 1 mM βME. Centrifuge-cleared lysate was applied to a Ni-NTA column (GE Healthcare), washed with a gradient from 50 mM NaH$_2$PO$_4$, 10 mM imidazole (pH 7.5), 150 mM NaCl, and 1 mM βME to 50 mM NaH$_2$PO$_4$, 50 mM imidazole (pH 7.5), 300 mM NaCl, and 1 mM βME. MBP-His$_{10}$-Ezrin(ABD) was eluted using a gradient of 10 mM→500 mM imidazole (pH 7.5) in 50 mM NaH$_2$PO$_4$, 150 mM NaCl, and 1 mM βME. Protein was concentrated using Amicon Ultra Centrifugal Filter units (Millipore) and MBP was cleaved by Factor Xa treatment for 10 hr at 4°C. Cleaved protein was further purified by size exclusion chromatography using a Superdex 75 10/300 GL column in 50 mM Tris-HCl (pH 7.5), 300 mM NaCl, 1 mM DTT, and 10% glycerol.

### Grb2 purification

BL21(DE3) cells containing GST- Grb2 were collected by centrifugation and lysed by sonication in 25 mM Tris-HCl (pH 8.0), 200 mM NaCl, 2 mM EDTA (pH 8.0), 5 mM βME, 1 mM PMSF, 1 µg/ml anti-pain, 1 µg/ml pepstatin, and 1 µg/ml leupeptin. Centrifuge-cleared lysate was applied to Glutathione Sepharose 4B (GE Healthcare) and washed with 25 mM Tris-HCl (pH 8.0), 200 mM NaCl, and 1 mM DTT. GST was cleaved from protein by TEV protease treatment for 16 hr at 4°C. Cleaved protein was applied to a Source 15 Q anion exchange column and eluted with a gradient of 0 mM → 300 mM NaCl in 20 mM imidazole (pH 7.0) and 1 mM DTT followed by size exclusion chromatography using a Superdex 75 prepgrade column (GE Healthcare) in 25 mM HEPES (pH 7.5), 150 mM NaCl, 1 mM MgCl$_2$, 1 mM βME, and 10% glycerol.

### Grb2$_{Basic}$ purification

BL21(DE3) cells containing GST-Grb2$_{Basic}$ were collected by centrifugation and lysed by sonication in 25 mM Tris-HCl (pH 8.0), 200 mM NaCl, 2 mM EDTA (pH 8.0), 5 mM βME, 1 mM PMSF, 1 µg/ml antipain, 1 µg/ml pepstatin, and 1 µg/ml leupeptin. Centrifuge-cleared lysate was applied to Glutathione Sepharose 4B (GE Healthcare) and washed with 25 mM Tris-HCl (pH 8.0), 200 mM NaCl, and 1 mM DTT. GST was cleaved from protein by TEV protease treatment for 16 hr at 4°C.

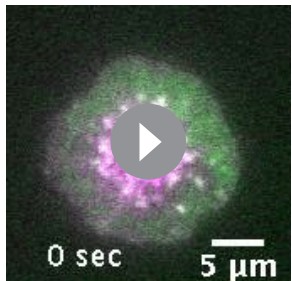

**Video 12.** Nck is maintained in LAT condensates as they move across the IS in activated Jurkat T cells treated with the formin inhibitor SMIFH2. TIRF microscopy of an activated Jurkat T cell expressing LAT-mCherry (magenta) and Nck-sfGFP (green) on a SLB coated with ICAM-1 and OKT3 revealed that Nck is maintained in LAT condensates as they move from the synapse of the cell to the cSMAC following treatment with SMIFH2 for 5 min prior to activation. Movie shows a 27 µm x 27 µm field of view. The movie is played at 10 fps with frame intervals of 5 s.
DOI: https://doi.org/10.7554/eLife.42695.050

**Table 1.** Sequences of constructs used in the study.

| Construct | Sequence | Notes |
|---|---|---|
| Ezrin | ISHMHHHHHHHHHKCKVYEPVSYHVQESLQDEGAEP TGYSAELSSEGIRDDRNEEKRITEAEKNERVQRQLLTLSSELSQARD ENKRTHNDIIHNENMRQGRDKYKTLRQIRQGNTKQRIDEFEAL | Human, Ezrin actin binding domain, residues 477–586, with N-terminal $His_{10}$ - KCK fusion. |
| Grb2 | GPLGSMEAIAKYDFKATADDELSFKRGDILKVLNEECDQN WYKAELNGKDGFIPKNYIEMKPHPWFFGKIPRAKAEEMLSKQRHDGAFLIRESESA PGDFSLSVKFGNDVQHFKVLRDGAGKYFLWVVKFNSLNELVDYHRSTSVSRNQQIF LR DIEQVPQQPTYVQALFDFDPQEDGELGFRRGDFIHVMDNSDPN WWKGACHGQTGMFPRNYVTPVNRNV | Human, residues 1–217. |
| Grb2<sub>Basic</sub> | GPLGSMEAIAKYDFKATADDELSFKRGDILKVLNEECDQNWY KAELNGKDGFIPKNYIEMKPHPWFFGKIPRAKAEEMLSKQRHDGAFLIRESESAPG DFSLSVKFGNDVQHFKVLRDGAGKYFLWVVKFNSLNELVDYHRSTSVSRNQQIF LRDIEQ VPQQPTYVQALFDFDPQEDGELGFRRGDFIHVMDNSDPNWWKGACHGQTGMF PRNY VTPVNRNVKEKKKGKAKKKRLTKGKEKKKGKAKKKRITKA | Human, residues 1–217 with the basic region of N-WASP (x2) fused to the C-terminal. |
| LAT | GGSLEHHHHHHHHGIQFKRPHTVAPWPPAFPPVTSFPPLSQPD LLPIPRSPQPLGGSHRTPSSRRDSDGANSVASFENEEPACEDADEDEDDFHNPGYL VVLPDSTPATSTAAPSAPALSTPGIRDSAFSMESIDDYVNVPESGESAEAS LDGSREYVNVSQELHPGAAKTEPAALSSQEAEEVEEEGAPDYENLQELN | Human, residues 48–233 (short isoform) with $His_8$ N-terminal fusion. This construct only contains the four C-terminal Tyr residues (Y132, Y171, Y191, and Y226) that are sufficient for TCR signaling. Y171, Y191, and Y226 are the three tyrosines that are recognized by Grb2 when phosphorylated. All other tyrosine residues were mutated to Phe residues. |
| Nck | GHMCMAEEVVVAKFDYVAQQEQELDIKKNERLWLLDDSKSWWRV RNSMNKTGFVPSNYVERKNSARKASIVKNLKDTLGIGKVKRKPSVPDSASPADDSFV DPGERLYDLNMPAYVKFNYMAEREDELSLIKGTKVIVMEKSSDGWWRGSYNGQVG WFPSNYVTEEGDSPLGDHVGSLSEKLAAVVNNLNTGQVLHVVQALYPFSSSNDEE LNFEKGDVMDVIEKPENDPEWWKARKINGMVGLVPKNYVTVMQNNPLTSGLEPS PPQSDYIRPSLTGKFAGNPWYYGKVTRHQAEMALNERGHEGDFLIRDSESSPNDF SVSLKAQGKNKHFKVQLKETVYSIGQRKFSTMEELVEHYKKAPIFTSEQGEKLYLVKH LS | Human, residues 1–377, with mutations C139S, C232A, C266S, C340S. A single Cys was added to the N-terminus. |
| His-Nck (FL) | GHMHHHHHHHHGGSCGGSGGSGGSGGSLEMAEEVVVAKFDYVAQQEQELD IKKNERLWLLDDSKSWWRVRNSMNKTGFVPSNYVERKNSARKASIVKNLKDT LGIGKVKRKPSVPDSASPADDSFVDPGERLYDLNMPAYVKFNYMAEREDELSL IKGTKVIVMEKSSDGWWRGSYNGQVGWFPSNYVTEEGDSPLGDHVGSLSEK LAAVVNNLNTGQVLHVVQALYPFSSSNDEELNFEKGDVMDVIEKPENDPEW WKARKINGMVGLVPKNYVTVMQNNPLTSGLEPSPPQSDYIRPSLTGKFAGNP WYYGKVTRHQAEMALNERGHEGDFLIRDSESSPNDFSVSLKAQGKNKHFK VQLKETVYSIGQRKFSTMEELVEHYKKAPIFTSEQGEKLYLVKHLS | Human, residues 1–377, with N-terminal $His_8$-GGSC(GGS)$_4$ fusion and C139S, C232A, C266S, and C340S mutations. |
| His-Nck (FLΔL1) | GHMHHHHHHHHGGSCGGSGGSGGSGGSLEAEEVVVAKFDYVAQQEQ ELDIKKNERLWLLDDSKSWWRVRNSMNKTGFVPSNYVERKNSGGSAGG SAGGSATLGIGKVKRKPSVPDSASPADDSFVDPGERLYDLNMPAYVKFNY MAEREDELSLIKGTKVIVMEKSSDGWWRGSYNGQVGWFPSNYVTEEGDS PLGDHVGSLSEKLAAVVNNLNTGQVLHVVQALYPFSSSNDEELNFEKGD VMDVIEKPENDPEWWKARKINGMVGLVPKNYVTVMQNNPLTSGLEPSPP QSDYIRPSLTGKFAGNPWYYGKVTRHQAEMALNERGHEGDFLIRDSESSP NDFSVSLKAQGKNKHFKVQLKETVYSIGQRKFSTMEELV EHYKKAPIFTSEQGEKLYLVKHLS | Human, residues 2–377, with N-terminal $His_8$-GGSC(GGS)$_4$ fusion and C139S, C232A, C266S, and C340S mutations. ARKASIVKNLKD in the N-terminal portion of L1 has been replaced with GGSAGGSAGGSA. |
| His-Nck (L1-S2-L2-S3-L3-SH2) | GHMHHHHHHHHGGSCGGSGGSGGSGGSLEKNSARKASIVKNLKDTLGIGK VKRKPSVPDSASPADDSFVDPGERLYDLNMPAYVKFNYMAEREDELSLIKGT KVIVMEKSSDGWWRGSYNGQVGWFPSNYVTEEGDSPLGDHVGSLSEKLAA VVNNLNTGQVLHVVQALYPFSSSNDEELNFEKGDVMDVIEKPENDPEWWKC RKINGMVGLVPKNYVTVMQNNPLTSGLEPSPPQSDYIRPSLTGKFAGNPWYY GKVTRHQAEMALNERGHEGDFLIRDSESSPNDFSVSLKAQGKNKHFKVQL KETVYSIGQRKFSTMEELVEHYKKAPIFTSEQGEKLYLVKHLS | Human, residues 59–377 (No S1), with N-terminal $His_8$-GGSC(GGS)$_4$ fusion and C139S, C232A, C266S, and C340S mutations. |

*Table 1 continued on next page*

*Table 1 continued*

| Construct | Sequence | Notes |
|---|---|---|
| His-Nck (L1(KtoE)-S2-L2-S3-L3-SH2) | GMHHHHHHHHGGSCGGSGGSGGSGGSLEKNSAREASIVENLEDTLGIG KVKRKPSVPDSASPADDSFVDPGERLYDLNMPAYVKFNYMAEREDELSLIKG TKVIVMEKSSDGWWRGSYNGQVGWFPSNYVTEEGDSPLGDHVGSLSEKLAA VVNNLNTGQVLHVVQALYPFSSSNDEELNFEKGDVMDVIEKPENDPEWWKC RKINGMVGLVPKNYVTVMQNNPLTSGLEPSPPQSDYIRPSLTGKFAGNPWYY GKVTRHQAEMALNERGHEGDFLIRDSESSPNDFSVSLKAQGKNKHFKVQLK ETVYSIGQRKFSTMEELVEHYKKAPIFTSEQGEKLYLVKHLS | Human, residues 59–377 (No S1), with N-terminal His$_8$-GGSC(GGS)$_4$ fusion and C139S, C232A, C266S, and C340S mutations. The basic N-terminal portion of L1 has the following basic to acidic mutations: K64E, K69E, and K72E. |
| His-Nck (L1(HM)-S2-L2-S3-L3-SH2) | GMHHHHHHHHGGSCGGSGGSGGSGGSLEKNSARKASNSKNSKDTLGIGKVK RK PSVPDSASPADDSFVDPGERLYDLNMPAYVKFNYMAEREDELSLIKGTKVIVMEK SSD GWWRGSYNGQVGWFPSNYVTEEGDSPLGDHVGSLSEKLAAVVNNLNTGQVLH VVQ ALYPFSSSNDEELNFEKGDVMDVIEKPENDPEWWKCRKINGMVGLVPK NYVTVMQN NPLTSGLEPSPPQSDYIRPSLTGKFAGNPWYYGKVTRHQAEMALNERGHEGDFLI RD SESSPNDFSVSLKAQGKNKHFKVQLKETVYSIGQRKFST MEELVEHYKKAPIFTSEQGEKLYLVKHLS | Human, residues 59–377, with N-terminal His$_8$-GGSC(GGS)$_4$ fusion and C139S, C232A, C266S, and C340S mutations. Additional Hydrophobic Mutations in L1: I67N, V68S, L71S. |
| His-Nck (S1-L1-S3-L3-SH2) | GMHHHHHHHHGGSCGGSGGSGGSGGSLEAEEVVVAKFDYVAQQEQELDIKK NERLW LLDDSKSWWRVRNSMNKTGFVPSNYVERKNSARKASIVKNLKDTLGIGKVKRK PSVPDSA SPADDSFVDPGERLYDLNVLHVVQALYPFSSSNDEELNFEKGDVMDVIEKPENDPE WWK CRKINGMVGLVPKNYVTVMQNNPLTSGLEPSPPQSDYIRPSLTGKFAGNPWYYGK VTRH QAEMALNERGHEGDFLIRDSESSPNDFSVSLKAQGKNKHFKVQLKETVYSIGQRKF STM EELVEHYKKAPIFTSEQGEKLYLVKHLS | Human, residues 2–108 and 191–377, with N-terminal His$_8$-GGSC (GGS)$_4$ fusion and C232A, C266S, and C340S mutations. |
| His-Nck (S1-L1-S2-L2-S3) | GMHHHHHHHHGGSCGGSGGSGGSGGSLEMAEEVVVAKFDYVAQQEQE LDIKKNERL WLLDDSKSWWRVRNSMNKTGFVPSNYVERKNSARKASIVKNLKDTLGIGKVKRK PSVPD SASPADDSFVDPGERLYDLNMPAYVKFNYMAEREDELSLIKGTKVIVMEKSSDG WWRGSYN GQVGWFPSNYVTEEGDSPLGDHVGSLSEKLAAVVNNLNTGQVLHVVQALYPF SSSNDE ELNFEKGDVMDVIEKPENDPEWWKARKINGMVGLVPKNYVTVMQN | Human, residues 2–252, with N-terminal His$_8$-GGSC(GGS)$_4$ fusion and C139S and C232A mutations. |
| His-Nck (S2-L2-S3-L3-SH2) | GMHHHHHHHHGGSCGGSGGSGGSGGSLEDLNMPAYVKFNYMAEREDE LSLIKGTKVI VMEKSSDGWWRGSYNGQVGWFPSNYVTEEGDSPLGDHVGSLSEKLAA VVNNLNTGQV LHVVQALYPFSSSNDEELNFEKGDVMDVIEKPENDPEWWKCRKINGMVGLVPK NYVTVM QNNPLTSGLEPSPPQSDYIRPSLTGKFAGNPWYYGKVTRHQAEMALNERGXEGXF LIRDS ESSPNDFSVSLKAQGKNKHFKVQLKETVYSIGQRKFSTMEELVEHYKKAPIFTSE QGEKLYLVKHLS | Human, residues 106–377, with N-terminal His$_8$-GGSC(GGS)$_4$ fusion and C139S, C232A, C266S, and C340S mutations. |
| His-Nck (S3-L3-SH2) | GMHHHHHHHHGGSCGGSGGSGGSGGSLEQVLHVVQALYPFSSSNDEE LNFEKGDVM DVIEKPENDPEWWKARKINGMVGLVPKNYVTVMQNNPLTSGLEPSPPQSDYI RPSLTG KFAGNPWYYGKVTRHQAEMALNERGHEGDFLIRDSESSPNDFSVSLKAQGK NKHFKV QLKETVYSIGQRKFSTMEELVEHYKKAPIFTSEQGEKLYLVKHLS | Human, residues 190–377, with N-terminal His$_8$-GGSC(GGS)$_4$ fusion and C232A, C266S, and C340S mutations. |
| His-Nck (L3-SH2) | GMHHHHHHHHGGSCGGSGGSGGSGGSLENPLTSGLEPSPPQSDYI RPSLTGKFAGNPW YYGKVTRHQAEMALNERGHEGDFLIRDSESSPNDFSVSLKAQGKNKHFKVQLKE TVYSIG QRKFSTMEELVEHYKKAPIFTSEQGEKLYLVKHLS | Human, residues 253–377, with N-terminal His$_8$-GGSC(GGS)$_4$ fusion and C266S and C340S mutations. |
| His-Nck (S1) | GMHHHHHHHHGGSCGGSGGSGGSGGSLEAEEVVVAKFDYVAQQEQELD IKKNERLWLLDDSKSWWRVRNSMNKTGFVPSNYVERKNSGSAAAS | Human, residues 2–61, with N-terminal His$_8$-GGSC(GGS)$_4$ fusion. |

*Table 1 continued on next page*

Table 1 continued

| Construct | Sequence | Notes |
|---|---|---|
| His-Nck (S2) | GHMHHHHHHHHGGSCGGSGGSGGSGGSLEDLNMPAYVKFNYMAEREDELSL IKGTKVIVMEKSSDGWWRGSYNGQVGWFPSNYVTEEGDGSAAAS | Human, residues 106–165, with N-terminal His$_8$-GGSC(GGS)$_4$ fusion and C139S mutation. |
| His-Nck (S3) | GHMHHHHHHHHGGSCGGSGGSGGSGGSLEQVLHVVQALYPFSSSNDE ELNFEKGDVMDVIEKPENDPEWWKARKINGMVGLVPKNYVTVMQNGSAAAS | Human, residues 190–252, with N-terminal His$_8$-GGSC(GGS)$_4$ fusion and C232A mutation. |
| His-Nck$_{Acidic}$ (FL(L1 KtoE)) | GHMHHHHHHHHGGSCGGSGGSGGSGGSLEAEEVVVAKFDYVAQQEQELDIK KNERLWLLDDSKSWWRVRNSMNKTGFVPSNYVERKNSAREASIVENLEDTLGIG KVKRKPSVPDSASPADDSFVDPGERLYDLNMPAYVKFNYMAEREDELSLIKGTKV IVMEKSSDGWWRGSYNGQVGWFPSNYVTEEGDSPLGDHVGSLSEKLAAVVNNL NTGQVLHVVQALYPFSSSNDEELNFEKGDVMDVIEKPENDPEWWKARKINGMV GLVPKNYVTVMQNNPLTSGLEPSPPQSDYIRPSLTGKFAGNPWYYGKVTRHQAE MALNERGHEGDFLIRDSESSPNDFSVSLKAQGKNKHFKVQLKETVYSIGQRKF STMEELVEHYKKAPIFTSEQGEKLYLVKHLS | Human, residues 2–377, with C139S, C232A, C266S, and C340S mutations. The basic N-terminal portion of L1 has the following basic to acidic mutations: K64E, K69E, and K72E. |
| His-Nck$_{Basic}$ (FL(L1 basic)) | GHMHHHHHHHHGGSCGGSGGSGGSGGSLEAEEVVVAKFDYVAQQEQELDIK KNERLWLLDDSKSWWRVRNSMNKTGFVPSNYVERKNSARKASIVKNLKDTLGI GKVKRKPSVPKSASPADKSFVKPGKRLYKLNMPAYVKFNYMAEREDELSLIKGTK VIVMEKSSDGWWRGSYNGQVGWFPSNYVTEEGDSPLGDHVGSLSEKLAAVVN NLNTGQVLHVVQALYPFSSSNDEELNFEKGDVMDVIEKPENDPEWWKARKING MVGLVPKNYVTVMQNNPLTSGLEPSPPQSDYIRPSLTGKFAGNPWYYGKVTRH QAEMALNERGHEGDFLIRDSESSPNDFSVSLKAQGKNKHFKVQLKETVYSIGQ RKFSTMEELVEHYKKAPIFTSEQGEKLYLVKHLS | Human, residues 2–377, with C139S, C232A, C266S, and C340S mutations. The acidic C-terminal portion of L1 has the following acidic to basic mutations: D88K, D95K, D99K, E102K, and D106K. |
| His-Nck$_{Neutral}$ (FL(L1 GGSA)$_{10}$) | GHMHHHHHHHHGGSCGGSGGSGGSGGSLEMAEEVVVAKFDYVAQQEQELD IKKNERLWLLDDSKSWWRVRNSMNKTGFVPSNYVERGGSAGGSAGGSAGGSA GGSAGSGGSAGGSAGGSAGGSAGGSAGSDLNMPAYVKFNYMAEREDELSLIKG TKVIVMEKCSDGWWRGSYNGQVGWFPSNYVTEEGDSPLGDHVGSLSEKLAAV NNLNTGQVLHVVQALYPFSSSNDEELNFEKGDVMDVIEKPENDPEWWKCRKIN GMVGLVPKNYVTVMQNNPLTSGLEPSPPQCDYIRPSLTGKFAGNPWYYGKVTRH QAEMALNERGHEGDFLIRDSESSPNDFSVSLKAQGKNKHFKVQLKETVYCIGQ RKFSTMEELVEHYKKAPIFTSEQGEKLYLVKHLS | Human full-length Nck with L1 replaced by a GGSA linker, residues 1–58, 109–377, with an N-terminal His$_8$-GGSC (GGS)$_4$ fusion and C139S, C232A, C266S, and C340S mutations. |
| N-WASP | GSESRFYFHPISDLPPPEPYVQTTKSYPSKLARNESRENESLFTFLGKKCVTMSS AVVQLYAADRNCMWSKKCSGVACLVKDNPQRSYFLRIFDIKDGKLLWEQELYNN FVYNSPRGYFHTFAGDTCQVALNFANEEEAKKFRKAVTDLLGRRQRKSEKRRDPPN GPNLPMATVDIKNPEITTNRFYGPQVNNISHTKEKKKGKAKKKRLTKADIGTPSNFQ HIGHVGWDPNTGFDLNNLDPELKNLFDMCGISEAQLKDRETSKVIYDFIEKTGG VEA VKNELRRQAPPPPPSRGGPPPPPPPPHNSGPPPPPARGRGAPPPPPSRAPTAAP PPPPPSRPSVAVPPPPPNRMYPPPPPALPSSAPSGPPPPPPSVLGVGPVAPPPPPP PPPPPGPPPPPGLPSDGDHQVPTTAGNKAALLDQIREGAQLKKVEQNSRPVSCSG RD ALLDQIRQGIQLKSVADGQESTPPTPAPTSGIVGALME VMQKRSKAIHSSDEDEDEDDEEDFEDDDEWED | Human WIP N-WASP EVH1 binding motif, residues 451–485 fused to human N-WASP, residues 26–505. |
| His-N-WASP | MGHHHHHHDYDIPTTENLYFQGSESRFYFHPISDLPPPEPYVQTTKSYPSKLARNES RENESLFTFLGKKCVTMSSAVVQLYAADRNCMWSKKCSGVACLVKDNPQRSYFLRI FDIKDGKLLWEQELYNNFVYNSPRGYFHTFAGDTCQVALNFANEEEAKKFRKA VTD LLGRRQRKSEKRRDPPNGPNLPMATVDIKNPEITTNRFYGPQVNNISHTKEKKKGK AKKKRLTKADIGTPSNFQHIGHVGWDPNTGFDLNNLDPELKNLFDMCGISEA QLKD RETSKVIYDFIEKTGGVEAVKNELRRQAPPPPPSRGGPPPPPPPPHNSGPPPPPAR GRGAPPPPPSRAPTAAPPPPPPSRPSVAVPPPPPNRMYPPPPPALPSSAPSGPPPP PPSVLGVGPVAPPPPPPPPPPPGPPPPPGLPSDGDHQVPTTAGNKAALLDQIREGA QLKKVEQNSRPVSCSGRDALLDQIRQGIQLKSVADGQESTPPTPAPTSGIVGALME VMQKRSKAIHSSDEDEDEDDEEDFEDDDEWED | Human WIP N-WASP EVH1 binding motif, residues 451–485 fused to human N-WASP, residues 26–505 with N-terminal His$_6$ fusion. |

Table 1 continued on next page

*Table 1 continued*

| Construct | Sequence | Notes |
|---|---|---|
| N-WASP_Basic | GSESRFYFHPISDLPPPEPYVQTTKSYPSKLARNESRENESLFTFLGKKCVTMSSA VVQ LYAADRNCMWSKKCSGVACLVKDNPQRSYFLRIFDIKDGKLLWEQELYNNF VYNSPRG YFHTFAGDTCQVALNFANEEEAKKFRKAVTDLLGRRQRKSEKRRDPPNGPNLPMA TV DIKNPEITTNRFYGPQVNNISHTKEKKKGKAKKKRLTKGKEKKKGKAKKKRITKADIG T PSNFQHIGHVGWDPNTGFDLNNLDPELKNLFDMCGISEAQLKDRETSKVIYDFIEK TG GVEAVKNELRRQAPPPPPSRGGPPPPPPPHNSGPPPPPARGRGAPPPPPSRAPT AAPPPPPPSRPSVAVPPPPPNRMYPPPPPALPSSAPSGPPPPPPSVLGVGPVAPPPP PPPPPPPGPPPPPGLPSDGDHQVPTTAGNKAALLDQIREGAQLKKVEQNSRPVSCS GRDALLDQIRQGIQLKSVADGQESTPPTPAPTSGIVGALMEVMQKRSKAIH SSDEDE DEDDEEDFEDDDEWED | Human WIP N-WASP EVH1 binding motif, residues 451–485 fused to human N-WASP, residues 26–505. Basic Region (186-200) has been doubled. |
| His-N-WASP_Basic | MGHHHHHHDYDIPTTENLYFQGSESRFYFHPISDLPPPEPYVQTTKSYPSKLARNE SR ENESLFTFLGKKCVTMSSAVVQLYAADRNCMWSKKCSGVACLVKDNPQRSYF LRIFDIK DGKLLWEQELYNNFVYNSPRGYFHTFAGDTCQVALNFANEEEAKKFRKAVTDLLG RRQ RKSEKRRDPPNGPNLPMATVDIKNPEITTNRFYGPQVNNISHTKEKKKGKAKKK RLTKG KEKKKGKAKKKRITKADIGTPSNFQHIGHVGWDPNTGFDLNNLDPELKNLFD MCGISEA QLKDRETSKVIYDFIEKTGGVEAVKNELRRQAPPPPPSRGGPPPPPPPHNSG PPPPP ARGRGAPPPPPSRAPTAAPPPPPPSRPSVAVPPPPPNRMYPPPPPALPSSAPSG PPPPP PSVLGVGPVAPPPPPPPPPPPGPPPPPGLPSDGDHQVPTTAGNKAALLDQIREGA QLKK VEQNSRPVSCSGRDALLDQIRQGIQLKSVADGQESTPPTPAPTSGIVGALMEVMQK RSKAIH SSDEDEDEDDEEDFEDDDEWED | Human WIP N-WASP EVH1 binding motif, residues 451–485 fused to human N-WASP, residues 26–505 with N-terminal His$_6$ fusion. Basic Region (186-200) has been doubled. |
| N-WASP_Neutral | GSESRFYFHPISDLPPPEPYVQTTKSYPSKLARNESRENESLFTFLGKKCVTMSSA VVQLYA ADRNCMWSKKCSGVACLVKDNPQRSYFLRIFDIKDGKLLWEQELYNNFVYNSPRG YFHTFA GDTCQVALNFANEEEAKKFRKAVTDLLGRRQRKSEKRRDPPNGPNLPMATVDIK NPEITT NRFYGPQVNNISHTGGSGGSGGSGGSGGSADIGTPSNFQHIGHVGWDPNTGFD LNNLD PELKNLFDMCGISEAQLKDRETSKVIYDFIEKTGGVEAVKNELRRQAPPPPPSRGG PPPPP PPPHNSGPPPPPARGRGAPPPPPSRAPTAAPPPPPPSRPSVAVPPPPPNRMYPPPPPA LP SSAPSGPPPPPPSVLGVGPVAPPPPPPPPPPPGPPPPPGLPSDGDHQVPTTAGNKAA LLDQ IREGAQLKKVEQNSRPVSCSGRDALLDQIRQGIQLKSVADGQESTPPTPAPTSGI VGALMEVM QKRSKAIHSSDEDEDEDDEEDFEDDDEWED | Human WIP N-WASP EVH1 binding motif, residues 451–485 fused to human N-WASP, residues 26–505. Basic Region (186-200) has been removed and replaced with (GGS)$_5$. |

*Table 1 continued on next page*

Table 1 continued

| Construct | Sequence | Notes |
|---|---|---|
| His-N-WASP<sub>Neutral</sub> | MGHHHHHHDYDIPTTENLYFQGSESRFYFHPISDLPPPEPYVQTTKSYPSKLARNESRE NESLFTFLGKKCVTMSSAVVQLYAADRNCMWSKKCSGVACLVKDNPQRSYFLRIFDIKDG KLLWEQELYNNFVYNSPRGYFHTFAGDTCQVALNFANEEEAKKFRKAVTDLLGRRQRKS EKRRDPPNGPNLPMATVDIKNPEITTNRFYGPQVNNISHTGGSGGSGGSGGSGGSADI GTPSNFQHIGHVGWDPNTGFDLNNLDPELKNLFDMCGISEAQLKDRETSKVIYDFIEKTG GVEAVKNELRRQAPPPPPPSRGGPPPPPPPPHNSGPPPPPARGRGAPPPPPSRAPTAA PPPPPPSRPSVAVPPPPPNRMYPPPPPALPSSAPSGPPPPPPSVLGVGPVAPPPPPPPP PPPGPPPPPGLPSDGDHQVPTTAGNKAALLDQIREGAQLKKVEQNSRPVSCSGRDALLD QIRQGIQLKSVADGQESTPPTPAPTSGIVGALMEVMQKRSKAIHS SDEDEDEDDEEDFEDDDEWED | Human WIP N-WASP EVH1 binding motif, residues 451–485 fused to human N-WASP, residues 26–505 with N-terminal His<sub>6</sub> fusion. Basic Region (186-200) has been removed and replaced with (GGS)<sub>5</sub>. |
| N-WASP BPVCA | GSEFKEKKKGKAKKKRAPPPPPPSRGGPPPPPPPPHSSGPPPPPARGRGA PPPPPSRA PTAAPPPPPPSRPGVVVPPPPPNRMYPHPPPALPSSAPSGPPPPPPLSMAGSTA PPPPP PPPPPPGPPPPPGLPSDGDHQVPASSGNKAALLDQIREGAQLKKVEQNSRPVSCSG RD ALLDQIRQGIQLKSVSDGQESTPPTPAPTSGIVGALMEVMQK RSKAIHSSDEDEDDDDEEDFEDDDEWED | Rat, N-WASP fusion of basic, proline-rich, and VCA domains, residues 183–193 and 273–501. |
| N-WASP BPVCA (R/A R/A) | GSEFKEKKKGKAKKKRAPPPPPPSRGGPPPPPPPPHSSGPPPPPARGRGA PPPPPSRAP TAAPPPPPPSRPGVVVPPPPPNRMYPHPPPALPSSAPSGPPPPPPLSMAGSTA PPPPP PPPPPGPPPPPGLPSDGDHQVPASSGNKAALLDQIAEGAQLKKVEQNSRPVSCSG RDAL LDQIAQGIQLKSVSDGQESTPPTPAPTSGIVGALMEVMQKRSKAIHS SDEDEDDDDEEDFEDDDEWED | Rat, N-WASP fusion of basic, proline-rich, and VCA domains, residues 183–193 and 273–501 with R410A and R438A mutations. |
| N-WASP BP | GSEFKEKKKGKAKKKRLTKADIGTPSNFQHIGHVGWDPN TGFDLNNLDPELKNLFDMCGIS APPPPPPSRGGPPPPPPPPHSSGPPPPPARGRGAPPPP PSRAPTAAPPPPPPSRPGVVVPP PPPNRMYPPPPPALPSSAPSGPPPPPPLSMAGSTAPPPP PPPPPPGPPPPPGLPSDGDHQVP | Rat, N-WASP fusion of basic, proline-rich, residues 183–239 and 273–396. |
| SLP-76 | GHMDNGGWSSFEEDDYESPNDDQDGEDDGDYESPNEEEEAPVEDDADYE PPPSNDEEAL QNSILPAKPFPNSNSMFIDRPPSGKTPQQPPVPPQRPMAALPPPPAGRNH SPLPPPQTNH EEPSRSRNHKTAKLPAPSIDRSTKPPLDRSLAPFDREPFTLGKKPPFSDKPSIPAG RSLGEHL PKIQKPPLPPTTERHERSSPLPGKKPPVPKHGWGPDRRENDEDDVHQRPLPQPA LLPMSSN TFPSRSTKPSPMNPLSSHMPGAFSESNSSFPQSASLPPFFSQGPSNRPPIRAEGRNF PLPL PNKPRPPSPAEEENCSLNEGSLEVLFQ | Human, residues 101–420, with a Cys added near the C-terminus for labeling. |
| Sos1 | GPLGSNDTVFIQVTLPHGPRSASVSSISLTKGTDEVPVPPPVPPRRRPESAPAESSPSKI MS KHLDSPPAIPPRQPTSKAYSPRYSISDRTSISDPPESPLLPPREPVRTPDVFSSSPLH LQPPP LGKKSDHGNAFFPNSPSPFTPPPPQTPSPHGTRRHLPSPPLTQEVDLHSIAG PPVPPRQSTSQ HIPKLPPKTYKREHTHPSMC | Human, poly-proline rich region, residues 1117–1319 with a Cys added at the C-terminus for labeling. |

Table 1 continued on next page

Table 1 continued

| Construct | Sequence | Notes |
|---|---|---|
| WASP | GSESRFYFHPISDLPPPEPYVQTTKSYPSKLARNESRENESLFTFLGRKCLTLATA VVQLYL ALPPGAEHWTKEHCGAVCFVKDNPQKSYFIRLYGLQAGRLLWEQE LYSQLVYSTPTPFFHTFA GDDCQAGLNFADEDEAQAFRALVQEKIQKRNQRQSGDRRQLPPPPTPANEE RRGGLPPLPL HPGGDQGGPPVGPLSLGLATVDIQNPDITSSRYRGLPAPGPSPADKKRSGKKKI SKADIGAPS GFKHVSHVGWDPQNGFDVNNLDPDLRSLFSRAGISEAQLTDAETSKLIYDFIED QGGLEAVRQ EMRRQEPLPPPPPPSRGGNQLPRPPIVGGNKGRSGPLPPVPLGIAPPPPTPRG PPPPGRGG PPPPPPPATGRSGPLPPPPGAGGPPMPPPPPPPPPPPSSGNGPAPPPLPPALV PAGGLAPGGGRGALLDQIRQGIQLNKTPGAPESSALQPPPQ SSEGLVGALMHVMQKRSRAIHSSDEGEDQAGDEDEDDEWDD | Human WIP WASP EVH1 binding motif, residues 451–485 fused to human WASP, residues 39–502. |

DOI: https://doi.org/10.7554/eLife.42695.051

Cleaved protein was applied to a Source 15 s cation exchange column and eluted with a gradient of 500 mM → 1000 mM NaCl in 20 mM imidazole (pH 7.0) and 1 mM DTT followed by buffer exchange using a HiTrap 26/10 Desalting column (GE Healthcare) in 50 mM HEPES (pH 7.5), 250 mM NaCl, 2 mM βME, and 10% glycerol.

## LAT purification and modification

BL21(DE3) cells containing MBP-His$_8$-LAT 48–233-His$_6$ were collected by centrifugation and lysed by cell disruption (Emulsiflex-C5, Avestin) in 20 mM imidazole (pH 8.0), 150 mM NaCl, 5 mM βME, 0.1% NP-40, 10% glycerol, 1 mM PMSF, 1 µg/ml antipain, 1 µg/ml pepstatin, and 1 µg/ml leupeptin. Centrifugation-cleared lysate was applied to Ni-NTA agarose (Qiagen), washed with 10 mM imidazole (pH 8.0), 150 mM NaCl, 5 mM βME, 0.01% NP-40, and 10% glycerol, and eluted with 500 mM imidazole (pH 8.0), 150 mM NaCl, 5 mM βME, 0.01% NP-40, and 10% glycerol. The MBP tag and His$_6$ tag were removed using TEV protease treatment for 16 hr at 4°C. Cleaved protein was applied to a Source 15 Q anion exchange column and eluted with a gradient of 200 mM→300 mM NaCl in 20 mM HEPES (pH 7.0) and 2 mM DTT followed by size exclusion chromatography using a Superdex 200 prepgrade column (GE Healthcare) in 25 mM HEPES (pH 7.5), 150 mM NaCl, 1 mM MgCl$_2$, and 1 mM DTT. LAT was concentrated using Amicon Ultra Centrifugal Filter units (Millipore) to >400 µM, mixed with 25 mM HEPES (pH 7.5), 150 mM NaCl, 15 mM ATP, 20 mM MgCl$_2$, 2 mM DTT, and active GST-ZAP70 (SignalChem), and incubated for 24 hr at 30°C. Phosphorylated LAT (pLAT) was resolved on a Mono Q anion exchange column using a shallow 250 mM → 320 mM NaCl gradient in 25 mM HEPES (pH 7.5), 1 mM MgCl$_2$, and 2 mM βME to separate differentially phosphorylated species of LAT. Complete LAT phosphorylation was confirmed by mass spectrometry. pLAT was then exchanged into 25 mM HEPES (pH 7.0), 150 mM NaCl, and 1 mM EDTA (pH 8.0) using a HiTrap Desalting Column (GE Healthcare). C$_5$-maleimide Alexa488 was added in excess to pLAT in reducing agent-free buffer and incubated for 16 hr at 4°C. Following the incubation 5 mM βME was added to the labeling solution to quench the reaction. Excess dye was removed from Alexa488-labeled pLAT by size exclusion chromatography in 25 mM HEPES (pH 7.5), 150 NaCl, 1 mM MgCl$_2$, 1 mM βME, and 10% glycerol.

## Nck and Nck variant purification

BL21(DE3) cells containing GST-Nck1 (or Nck variant, both His- and non-His-tagged) were collected by centrifugation and lysed by sonication in 25 mM Tris-HCl (pH 8.0), 200 mM NaCl, 2 mM EDTA (pH 8.0), 1 mM DTT, 1 mM PMSF, 1 µg/ml antipain, 1 µg/ml pepstatin, and 1 µg/ml leupeptin. Centrifuge-cleared lysate was applied to Glutathione Sepharose 4B (GE Healthcare) and washed with 25 mM Tris-HCl (pH 8.0), 200 mM NaCl, and 1 mM DTT. GST was cleaved from protein by TEV protease treatment for 16 hr at 4°C. Cleaved protein was applied to a Source 15 Q anion exchange column and eluted with a gradient of 0 → 200 mM NaCl in 20 mM imidazole (pH 7.0) and 1 mM DTT. Eluted protein was pooled and applied to a Source 15 s cation exchange column and eluted with a gradient

of $0 \rightarrow 200$ mM NaCl in 20 mM imidazole (pH 7.0) and 1 mM DTT. Eluted protein was concentrated using Amicon Ultra 10K concentrators and further purified by size exclusion chromatography using a Superdex 75 prepgrade column (GE Healthcare) in 25 mM HEPES (pH 7.5), 150 mM NaCl, 1 mM βME, and 10% glycerol.

## N-WASP and WASP purification

BL21(DE3) cells containing $His_6$-N-WASP (or N-WASP variants) or Rosetta 2(DE3) cells containing $His_6$-WASP were collected by centrifugation and lysed by cell disruption (Emulsiflex-C5, Avestin) in 20 mM imidazole (pH 7.0), 300 mM KCl, 5 mM βME, 0.01% NP-40, 1 mM PMSF, 1 µg/ml antipain, 1 mM benzamidine, and 1 µg/ml leupeptin. The cleared lysate was applied to Ni-NTA agarose (Qiagen), washed with 50 mM imidazole (pH 7.0), 300 mM KCl, 5 mM βME, and eluted with 300 mM imidazole (pH 7.0), 100 mM KCl, and 5 mM βME. The eluate was further purified over a Source 15 Q column using a gradient of $250 \rightarrow 450$ mM NaCl in 20 mM imidazole (pH 7.0), and 1 mM DTT. The $His_6$-tag was removed by TEV protease at 4°C for 16 hr (for His-tagged variants, no TEV treatment occurred). Cleaved N-WASP (or uncleaved for His-tagged variants) or cleaved WASP was then applied to a Source 15 s column using a gradient of $110 \rightarrow 410$ mM NaCl in 20 mM imidazole (pH 7.0), 1 mM DTT. Fractions containing N-WASP or WASP were concentrated using an Amicon Ultra 10K concentrator (Millipore) and further purified by size exclusion chromatography using a Superdex 200 prepgrade column (GE Healthcare) in 25 mM HEPES (pH 7.5), 150 mM KCl, 1 mM βME, and 10% glycerol.

## SLP-76 purification and modification

BL21(DE3) cells containing MBP-SLP-76 Acidic and Proline Rich region-$His_6$ were collected by centrifugation and lysed by cell disruption (Emulsiflex-C5, Avestin) in 20 mM imidazole (pH 8.0), 150 mM NaCl, 5 mM βME, 0.01% NP-40, 10% glycerol, 1 mM PMSF, 1 µg/ml antipain, 1 µg/ml pepstatin, and 1 µg/ml leupeptin. Centrifuge-cleared lysate was applied to Ni-NTA Agarose (Qiagen), washed first with 20 mM imidazole (pH 8.0), 150 mM NaCl, 5 mM βME, 0.01% NP-40, 10% glycerol, and 1 mM benzamidine, then washed with 50 mM imidazole (pH 8.0), 300 mM NaCl, 5 mM βME, 0.01% NP-40, 10% glycerol, 1 mM benzamidine, and eluted with 500 mM imidazole (pH 8.0), 150 mM NaCl, 5 mM βME, 0.01% NP-40, 10% glycerol, and 1 mM benzamidine. MBP cleaved by TEV protease treatment for 16 hr at 4°C or for 2 hr at room temperature. $His_6$ was concurrently cleaved by PreScission protease treatment for 16 hr at 4°C or for 2 hr at room temperature. Cleaved protein was applied to a Source 15 Q anion exchange column and eluted with a gradient of 200 mM→350 mM NaCl in 20 mM HEPES (pH 7.5) and 2 mM βME followed by size exclusion chromatography using a Superdex 200 prepgrade column (GE Healthcare) in 25 mM HEPES (pH 7.5), 150 mM NaCl, 1 mM $MgCl_2$, and 1 mM DTT. To phosphorylate SLP-76, Purified SLP-76 was incubated in 50 mM HEPES (pH 7.0), 150 mM NaCl, 20 mM $MgCl_2$, 15 mM ATP, 2 mM DTT, and 20 nM Active GST-ZAP70 (SignalChem). Phosphorylated SLP-76 was resolved on a Mono Q anion exchange column (GE Healthcare) using a shallow gradient from 300 mM $\rightarrow$ 400 mM in 25 mM HEPES (pH 7.5) and 1 mM βME to separate differentially phosphorylated species of SLP-76. The phosphorylation state of pSLP-76 was confirmed by mass spectrometry analysis. pSLP-76 was then exchanged into a buffer containing 25 mM HEPES (pH 7.5), 150 mM NaCl, 1 mM $MgCl_2$, 1 mM βME, and 10% glycerol over a size exclusion column (Superdex 200 prepgrade column).

## Sos1 purification

BL21(DE3) cells containing GST-Sos1 were collected by centrifugation and lysed by sonication in 25 mM Tris-HCl (pH 8.0), 200 mM NaCl, 2 mM EDTA (pH 8.0), 1 mM DTT, 1 mM PMSF, 1 µg/ml antipain, 1 µg/ml pepstatin, and 1 µg/ml leupeptin. Centrifuge-cleared lysate was applied to Glutathione Sepharose 4B (GE Healthcare) and washed with 50 mM HEPES (pH 7.0), 150 mM NaCl, 10% glycerol and 1 mM DTT. GST was cleaved from protein by PreScission protease treatment overnight at 4°C. Cleaved protein was purified by size exclusion chromatography using a Superdex 200 prepgrade column (GE Healthcare) in 50 mM HEPES (pH 7.5), 150 mM NaCl, 1 mM βME, and 10% glycerol.

## Small unilamellar vesicle preparation

Synthetic 1,2-dioleyl-*sn*-glycero-3-phosphocholine (POPC), 1,2-dioleoyl-*sn*-glycero-3-[(*N*-(5-amino-1-carboxypentyl)iminodiacetic acid)succinyl](nickel salt) (DGS-NTA-Ni), L-α-phosphatidylserine (Brain PS), 1,2-distearoyl-*sn*-glycero-3-phosphoethanolamine-N-[biotinyl(polyethylene glycol)−2000] (ammonium salt) (DSPE-PEG 2000 Biotin), and 1,2-dioleoyl-*sn*-glycero-3-phosphoethanolamine-N-[methoxy(polyethyleneglycol)−5000] (ammonium salt) (PEG5000 PE) were purchased from Avanti Polar Lipids. Phospholipids for reconstitution assays (93% POPC, 5% PS, 2% DGS-NTA-Ni and 0.1% PEG 5000 PE) or (98% POPC, 1% DSPE-PEG2000 Biotin, 1% DGS-NTA-Ni, and 0.1% PEG5000-PE) were dried under a stream of Argon, desiccated over 3 hr, and resuspended in PBS (pH 7.3). To promote the formation of small unilamellar vesicles (SUVs), the lipid solution was repeatedly frozen in liquid $N_2$ and thawed using a 37°C water bath until the solution cleared. Cleared SUV-containing solution was centrifuged at 33,500 g for 45 min at 4°C. Supernatant containing SUVs was collected and stored at 4°C covered with Argon.

## Steady-State reconstitution assays

These methods are adapted from previously published methods (*Köster et al., 2016*; *Su et al., 2017*; *Su et al., 2016*). Supported lipid bilayers (SLBs) containing 93% POPC, 5% PS, 2% DGS-NTA-Ni, and trace PEG5000-PE were formed in 96-well glass-bottomed plates (Matrical). Glass was washed with Hellmanex III (Hëlma Analytics) for 4 hr at 50°C, thoroughly rinsed with MilliQ $H_2O$, washed with 6M NaOH for 30 min at 45°C two times, and thoroughly rinsed with MilliQ $H_2O$ followed by equilibration with 50 mM HEPES (pH 7.3), 150 mM KCl, and 1 mM TCEP. SUVs were added to cleaned wells covered by 50 mM HEPES (pH 7.3), 150 mM KCl, and 1 mM TCEP and incubated for 1 hr at 40°C to allow SUVs to collapse on glass and fuse to form the SLB. SLBs were washed with 50 mM HEPES (pH 7.3), 150 mM KCl, and 1 mM TCEP to remove excess SUVs. SLBs were blocked with 50 mM HEPES (pH 7.3), 150 mM KCl, 1 mM TCEP, and 1 mg/mL BSA for 30 min at room temperature. 20 nM His8-pLAT-Alexa488 and 5 nM His10-Ezrin(ABD) were premixed and incubated with SLBs in 50 mM HEPES (pH 7.3), 150 mM KCl, 1 mM TCEP, and 1 mg/ml BSA for 30 min. SLBs were then washed with 50 mM HEPES (pH 7.3), 75 mM KCl, 1 mM $MgCl_2$, 1 mM TCEP, and 1 mg/ml BSA to remove unbound His-tagged proteins. Polymerized, 10% rhodamine-labeled actin was added to the SLB and allowed to bind to membrane-attached Ezrin for 15 min. 100 nM myosin II and 1 mM ATP was added to the membrane-bound actin network to induce steady-state movement of actin filaments by myosin II activity. Indicated amounts of soluble proteins were added to His-tagged protein- and actomyosin-bound SLBs and imaged using TIRF microscopy. Time-lapse images were captured every 15 s for 15 min. To obtain replicates for each time-lapse, 5 fields of view were captured by sequential, multi-point image acquisition. Microscopy experiments were performed in the presence of a glucose/glucose oxidase/catalase $O_2$-scavenging system. Identical image acquisition settings were used for each steady-state biochemical experiment. See *Table 2* for more information.

## Actomyosin contraction assays

These methods are adapted from previously published methods (*Köster et al., 2016*; *Su et al., 2017*; *Su et al., 2016*). Supported lipid bilayers (SLBs) containing 93% POPC, 5% PS, 2% DGS-NTA-Ni, and trace PEG 5000 PE were formed in 96-well glass-bottomed plates (Matrical). Glass was washed with Hellmanex III (Hëlma Analytics) for 4 hr at 50°C, thoroughly rinsed with MilliQ $H_2O$, washed with 6M NaOH for 30 min at 45°C two times, and thoroughly rinsed with MilliQ $H_2O$ followed by equilibration with 50 mM HEPES (pH 7.3), 150 mM KCl, and 1 mM TCEP. SUVs were added to cleaned wells covered by 50 mM HEPES (pH 7.3), 150 mM KCl, and 1 mM TCEP and incubated for 1 hr at 40°C to allow SUVs to collapse on glass and fuse to form the SLB. SLBs were washed with 50 mM HEPES (pH 7.3), 150 mM KCl, and 1 mM TCEP to remove excess SUVs. SLBs were blocked with 50 mM HEPES (pH 7.3), 150 mM KCl, 1 mM TCEP, and 1 mg/mL BSA for 30 min at room temperature. 20 nM His8-pLAT-Alexa488 and 5 nM His10-Ezrin(ABD) were premixed and incubated with SLBs in 50 mM HEPES (pH 7.3), 150 mM KCl, 1 mM TCEP, and 1 mg/ml BSA for 30 min. SLBs were then washed with 50 mM HEPES (pH 7.3), 50 mM KCl, 1 mM $MgCl_2$, 1 mM TCEP, and 1 mg/ml BSA to remove unbound His-tagged proteins. Polymerized, 10% rhodamine-labeled actin was added to the SLB and allowed to bind to membrane-attached Ezrin for 15 min. Indicated

**Table 2.** Experimental conditions used in steady state and actomyosin contraction assays.

50 mM HEPES pH 7.3, 1 mM MgCl$_2$, 1 mM TCEP, 1 mg / ml BSA, and 0.2 mg / ml glucose oxidase +0.035 mg / ml catalase +1 mM glucose (O 2 scavenging system) were included in every assay.

| Experiment | His$_8$-pLAT | His$_{10}$-Ezrin (ABD) | Grb2 | Sos1 | pSLP-76 | Nck | N-wasp | WASP | Actin | Myosin II | KCl | ATP |
|---|---|---|---|---|---|---|---|---|---|---|---|---|
| pLAT → Sos1 (Steady-State) | 20 nM (sol'n) 500 molecules /µm$^2$ | five nM (sol'n) ~100 molecules /µm$^2$ | 125 nM | 125 nM | 0 nM | 0 nM | 0 nM | 0 nM | 0 nM | 0 nM | 75 mM | 1 mM |
| pLAT → Sos1 +Actin (Steady-State) | 20 nM (sol'n) 500 molecules /µm$^2$ | five nM (sol'n) ~100 molecules /µm$^2$ | 125 nM | 125 nM | 0 nM | 0 nM | 0 nM | 0 nM | 250 nM | 0 nM | 75 mM | 1 mM |
| pLAT → Sos1 +Actin + Myosin (Steady-State) | 20 nM (sol'n) 500 molecules /µm$^2$ | five nM (sol'n) ~100 molecules /µm$^2$ | 125 nM | 125 nM | 0 nM | 0 nM | 0 nM | 0 nM | 250 nM | 100 nM | 75 mM | 1 mM |
| pLAT → pSLP-76 (Steady-State) | 20 nM (sol'n) 500 molecules /µm$^2$ | five nM (sol'n) ~100 molecules /µm$^2$ | 125 nM | 62.5 nM | 62.5 nM | 0 nM | 0 nM | 0 nM | 0 nM | 0 nM | 75 mM | 1 mM |
| pLAT → pSLP-76 +Actin (Steady-State) | 20 nM (sol'n) 500 molecules /µm$^2$ | five nM (sol'n) ~100 molecules /µm$^2$ | 125 nM | 62.5 nM | 62.5 nM | 0 nM | 0 nM | 0 nM | 250 nM | 0 nM | 75 mM | 1 mM |
| pLAT → pSLP-76 +Actin + Myosin (Steady-State) | 20 nM (sol'n) 500 molecules /µm$^2$ | five nM (sol'n) ~100 molecules /µm$^2$ | 125 nM | 62.5 nM | 62.5 nM | 0 nM | 0 nM | 0 nM | 250 nM | 100 nM | 75 mM | 1 mM |
| pLAT → Nck (Steady-State) | 20 nM (sol'n) 500 molecules /µm$^2$ | five nM (sol'n) ~100 molecules /µm$^2$ | 125 nM | 62.5 nM | 62.5 nM | 125 nM | 0 nM | 0 nM | 0 nM | 0 nM | 75 mM | 1 mM |
| pLAT → Nck +Actin (Steady-State) | 20 nM (sol'n) 500 molecules /µm$^2$ | five nM (sol'n) ~100 molecules /µm$^2$ | 125 nM | 62.5 nM | 62.5 nM | 125 nM | 0 nM | 0 nM | 250 nM | 0 nM | 75 mM | 1 mM |
| pLAT → Nck +Actin + Myosin (Steady-State) | 20 nM (sol'n) 500 molecules /µm$^2$ | five nM (sol'n) ~100 molecules /µm$^2$ | 125 nM | 62.5 nM | 62.5 nM | 125 nM | 0 nM | 0 nM | 250 nM | 100 nM | 75 mM | 1 mM |
| pLAT → N-WASP (Steady-State) | 20 nM (sol'n) 500 molecules /µm$^2$ | five nM (sol'n) ~100 molecules /µm$^2$ | 125 nM | 62.5 nM | 62.5 nM | 125 nM | 125 nM | 0 nM | 0 nM | 0 nM | 75 mM | 1 mM |
| pLAT → N-WASP +Actin (Steady-State) | 20 nM (sol'n) 500 molecules /µm$^2$ | five nM (sol'n) ~100 molecules /µm$^2$ | 125 nM | 62.5 nM | 62.5 nM | 125 nM | 125 nM | 0 nM | 250 nM | 0 nM | 75 mM | 1 mM |
| pLAT → N-WASP +Actin + Myosin (Steady-State) | 20 nM (sol'n) 500 molecules /µm$^2$ | five nM (sol'n) ~100 molecules /µm$^2$ | 125 nM | 62.5 nM | 62.5 nM | 125 nM | 125 nM | 0 nM | 250 nM | 100 nM | 75 mM | 1 mM |
| pLAT → Sos1 +Actin + Myosin (Contraction) For Grb2, Grb2$_{Basic}$ | 20 nM (sol'n) 500 molecules /µm$^2$ | five nM (sol'n) ~100 molecules /µm$^2$ | 125 nM | 125 nM | 0 nM | 0 nM | 0 nM | 0 nM | 250 nM | 100 nM | 50 mM | 0.05 mM |
| pLAT → N-WASP +Actin + Myosin (Contraction) For N-WASP$_{WT}$, N-WASP$_{Neutral}$, N-WASP$_{Basic}$ | 20 nM (sol'n) 500 molecules /µm$^2$ | five nM (sol'n) ~100 molecules /µm$^2$ | 125 nM | 62.5 nM | 62.5 nM | 125 nM | 125 nM | 0 nM | 250 nM | 100 nM | 50 mM | 0.05 mM |
| pLAT → WASP (Steady-State) | 20 nM (sol'n) 500 molecules /µm$^2$ | five nM (sol'n) ~100 molecules /µm$^2$ | 125 nM | 62.5 nM | 62.5 nM | 125 nM | 0 nM | 125 nM | 0 nM | 0 nM | 75 mM | 1 mM |
| pLAT → WASP +Actin (Steady-State) | 20 nM (sol'n) 500 molecules /µm$^2$ | five nM (sol'n) ~100 molecules /µm$^2$ | 125 nM | 62.5 nM | 62.5 nM | 125 nM | 0 nM | 125 nM | 250 nM | 0 nM | 75 mM | 1 mM |
| pLAT → WASP +Actin + Myosin (Steady-State) | 20 nM (sol'n) 500 molecules /µm$^2$ | five nM (sol'n) ~100 molecules /µm$^2$ | 125 nM | 62.5 nM | 62.5 nM | 125 nM | 0 nM | 125 nM | 250 nM | 100 nM | 75 mM | 1 mM |

*Table 2 continued on next page*

*Table 2 continued*

| Experiment | His$_8$-pLAT | His$_{10}$-Ezrin (ABD) | Grb2 | Sos1 | pSLP-76 | Nck | N-wasp | WASP | Actin | Myosin II | KCl | ATP |
|---|---|---|---|---|---|---|---|---|---|---|---|---|
| pLAT → WASP +Actin + Myosin (Contraction) | 20 nM (sol'n) 500 molecules /μm$^2$ | five nM (sol'n) ~100 molecules /μm$^2$ | 125 nM | 62.5 nM | 62.5 nM | 125 nM | 0 nM | 125 nM | 250 nM | 100 nM | 50 mM | 0.05 mM |

DOI: https://doi.org/10.7554/eLife.42695.052

amounts of soluble proteins were added to His-tagged protein- and actin-bound SLBs and incubated for 15 min to allow for the formation of phase-separated condensates. TIRF microscopy was then used to capture actin contraction and associated condensate movement when 100 nM myosin II was added to the membrane-bound actin network to induce contraction of actin filaments by myosin II activity. Time-lapse images were captured every 5 s for up to 10 min, until contraction was completed. To obtain replicates for each time-lapse, 5 fields of view were captured by sequential, multi-point image acquisition. Microscopy experiments were performed in the presence of a glucose/glucose oxidase/catalase O$_2$-scavenging system. Identical image acquisition settings were used for each actomyosin contraction biochemical experiment. See *Table 2* for more information. These experiments were especially sensitive to light-induced artifacts, so images of nearby regions were captured outside of the field of view after the completion of actin contraction to ensure that time-lapse images were representative of condensate and actin movement rather than the result of an artifact (*Figure 2—figure supplement 1*).

## Actin binding assays on SLBs

Supported lipid bilayers (SLBs) containing 93% POPC, 5% PS, 2% DGS-NTA-Ni, and trace PEG5000-PE were formed in 96-well glass-bottomed plates (Matrical). Glass was washed with Hellmanex III (Hëlma Analytics) for 4 hr at 50℃, thoroughly rinsed with MilliQ H$_2$O, washed with 6M NaOH for 30 min at 45℃ two times, and thoroughly rinsed with MilliQ H$_2$O followed by equilibration with 50 mM HEPES (pH 7.3), 150 mM KCl, and 1 mM TCEP. SUVs were added to cleaned wells covered by 50 mM HEPES (pH 7.3), 150 mM KCl, and 1 mM TCEP and incubated for 1 hr at 40℃ to allow SUVs to collapse on glass and fuse to form the SLB. SLBs were washed with 50 mM HEPES (pH 7.3), 150 mM KCl, and 1 mM TCEP to remove excess SUVs. SLBs were blocked with 50 mM HEPES (pH 7.3), 150 mM KCl, 1 mM TCEP, and 1 mg/mL BSA for 30 min at room temperature. Varied concentrations of His-tagged pLAT, Nck fragments, Nck full-length variants, or N-WASP variants were incubated with SLBs in 50 mM HEPES (pH 7.3), 150 mM KCl, 1 mM TCEP, and 1 mg/ml BSA for 30 min. SLBs were then washed with 50 mM HEPES (pH 7.3), 50 mM KCl, 1 mM MgCl$_2$, 1 mM TCEP, and 1 mg/ml BSA to remove unbound His-tagged proteins. If assays required pLAT condensate formation, components were added to pLAT-coated membrane to induce condensate formation for 15 min at room temperature. Polymerized, 10% rhodamine-labeled actin was added to the SLB and allowed to bind to membrane-bound proteins. TIRF microscopy was used to capture images. Microscopy experiments were performed in the presence of a glucose/glucose oxidase/catalase O$_2$-scavenging system. Identical image acquisition settings were used for each actin-binding biochemical experiment. Images were analyzed using ImageJ (FIJI). The same brightness and contrast were applied to images within the same panels. Camera background was subtracted before calculating mean fluorescence intensities. Data from image analysis within FIJI was analyzed and graphed using GraphPad Prism v.7.

## Actin Co-sedimentation assays

G-actin was allowed to polymerize for 1 hr at room temperature in a buffer containing 50 mM HEPES pH 7.3, 50 mM KCl, 1 mM MgCl$_2$, 1 mM EGTA pH 8.0, and 1 mM ATP. In the first set of co-sedimentation experiments (*Figure 3—figure supplement 3A*), 2 μM F-actin and 2 μM α-actinin, 2 μM Grb2, 2 μM Nck, or 2 μM N-WASP were incubated at room temperature for 1 hr. In the second set of experiments (*Figure 3—figure supplement 3B*), 1 μM F-Actin, and various combinations of 1 μM pLAT, 1 μM Grb2, 1 μM pSLP-76, 1 μM Nck, and 1 μM N-WASP were incubated at room temperature for 1 hr. Solutions containing F-actin and potential binding partners were centrifuged at

100,000 g for 20 min at 20°C (*Figure 3—figure supplement 4*). Supernatant and pellet components were analyzed using SDS-PAGE gels stained with Coomassie blue. The intensity of Nck and N-WASP bands in the supernatant and pellet (*Figure 3—figure supplement 3*) or the intensity of N-WASP bands in the supernatant and pellet (*Figure 3—figure supplement 4*) were measured using FIJI and plotted using GraphPad Prism v.7. The $K_A$ of N-WASP for F-Actin was calculated using the following equation (*Blin et al., 2008*):

$$\frac{Bound\ N-WASP}{Total\ N-WASP} = \frac{K_A(Accesible\ F-Actin)}{1 + K_A(Accessible\ F-Actin)}$$

The $K_D$ of N-WASP for F-Actin was calculated by taking the reciprocal of the $K_A$.

## Cell culture and transduction

To generate lentiviruses expressing labeled LAT condensate components, LAT was inserted into either the pHR-mCitrine-tWPRE or pHR-mCherry-tWPRE backbone vector, Grb2 and Grb2$_{Basic}$ were inserted into the pHR-mCherry-tWPRE backbone vector, Nck was inserted into the pHR-sfGFP-tWPRE backbone vector, and Gads and LifeAct were inserted into the pHR-BFP-tWPRE backbone vector. HEK293T cells were grown in DMEM supplemented with 10% FBS, 100 U/ml penicillin, and 100 µg/mL streptomycin, and transfected with these plasmids in combination with those expressing viral proteins using Lipofectamine 2000. Lentivirus was collected two days post-transfection.

Jurkat T cells were grown in RPMI-1640 supplemented with 10% FBS, 100 U/mL penicillin, and 100 µg/mL streptomycin. Lentiviral transduction was used to make WT Jurkat T cells (E6.1) stably expressing combinations of LAT, Grb2, Gads, Nck, Grb2$_{Basic}$, or LifeAct.

All cells used in this study were tested for mycoplasma contamination using the Lonza MycoAlert Mycoplasma Detection Kit and found to be mycoplasma-free.

## Activation of jurkat T cells

Supported lipid bilayers (SLBs) containing 98% POPC, 1% DSPE-PEG 2000 Biotin, 1% DGS-Ni-NTA, and trace PEG 5000 PE were formed in 96-well glass-bottomed plates (Matrical). Glass was washed with Hellmanex III (Hëlma Analytics) for 4 hr at 50°C, thoroughly rinsed with MilliQ H$_2$O, washed with 6M NaOH for 30 min at 45°C two times, and thoroughly rinsed with MilliQ H$_2$O followed by equilibration with PBS (pH 7.3). SUVs were added to cleaned wells covered by PBS (pH 7.3) and incubated for 1 hr at 40°C to allow SUVs to collapse on glass and fuse to form the SLB. SLBs were washed with PBS (pH 7.3) to remove excess SUVs. 1 µg / ml streptavidin-Alexa647 (Thermo Fisher Scientific) in PBS (pH 7.3) was added to the SLB and incubated at room temperature for 30 min. SLBs were thoroughly washed with PBS (7.3) to remove excess streptavidin-Alexa647. 5 µg / ml OKT3-Biotin (Thermo Fisher Scientific) in PBS (pH 7.3) was added to the SLB and incubated at room temperature for 30 min. SLBs were thoroughly washed with PBS (pH 7.3) to remove excess OKT3-Biotin. 1 µg / ml His-ICAM-1 (Thermo Fisher Scientific) in PBS (pH 7.3) was added to the SLB and incubated at room temperature for 30 min. SLBs were thoroughly washed with PBS (pH 7.3) to remove excess His-ICAM-1. PBS (pH 7.3) was exchanged to RPMI 1640 supplemented with 20 mM HEPES (pH 7.4) by washing SLBs three times. WT Jurkat T Cells expressing either LAT-mCitrine, Grb2-mCherry, and Gads-BFP, LAT-mCherry, Nck-sfGFP, and LifeAct-BFP, or LAT-mCitrine and Grb2$_{Basic}$-mCherry were washed with serum-free and phenol red-free RPMI 5X to minimize serum component contribution to Jurkat T cell activation on bilayers, incubated in serum-free and phenol red-free RPMI for at least 1 hr at 37°C following washes, and dropped onto the bilayers. Condensate mobility and composition was imaged by TIRF microscopy at 37°C. Images were captured every 5 s for up to 5 min. Identical image acquisition settings were used for each cellular experiment.

## DMSO or SMIFH2 treatment of jurkat T cells

Jurkat T cells expressing LAT-mCherry, Nck-sfGFP, and LifeAct-BFP were prepared as described above. Prior to TIRF microscopy imaging cells were incubated with DMSO or 5 µM SMIFH2 in DMSO for 5 min.

## TIRF microscopy

TIRF images were captured using a TIRF/iLAS2 TIRF/FRAP module (Biovision) mounted on a Leica DMI6000 microscope base equipped with a Hamamatsu ImagEMX2 EM-CCD camera with a 100 × 1.49 NA objective. Images were acquired using MetaMorph software. Experiments using TIRF microscopy are described in detail in the following methods sections: Steady State Reconstitution Assays, Actomyosin Contraction Assays, Actin Binding Assays on SLBs, and Activation of Jurkat T Cells.

## Polarization microscopy and analysis

We used a Nikon TE-2000E microscope with custom-built polarization optical systems on an optical table. The laser beam (25 mW, OBIS 561 nm, Coherent) was routed via custom optics and focused on the back focal plane of a 100 × 1.49 NA TIRF objective (Nikon). The laser beam was circularly polarized using a combination of a half wave plate and quarter wave plate (Meadowlark Optics). The circularly polarized beam was rotated at 300–400 Hz in the back focal plane of the objective to achieve total internal reflection at the specimen plane to achieve isotropic excitation within the focal plane. A quadrant imaging system, as described in *Mehta et al. (2016)* was used for instantaneous image capture along four polarization orientations at 45° increments (I0, I45, I90, I135). Time-lapse images were captured using an EMCCD camera iXon+ (Andor Technology) every 5 s for up to 10 min at 37°C. Micro – Manager (version 1.4.15) software was used to acquire images. A detailed description of the analysis can be found in *Mehta et al. (2016)* and *Nordenfelt et al. (2017)*. Briefly, the first ten frames following IS formation from each time-lapse image set were analyzed using custom code developed in MATLAB 2014a (*Mehta, 2015*). Calculated polarizations for each detected SiR-Actin speckle were visualized on time-lapse image panels using FIJI and plotted on a radial line from the synapse center to the synapse edge and determined to be within 45° of perpendicular or parallel form the synapse edge.

## Confocal microscopy of activated jurkat T cells

Confocal images were captured using a Yokogawa spinning disk (Biovision) mounted on a Leica DMI6000 microscope base equipped with a Hamamatsu ImagEMX2 EM-CCD camera with a 100 × 1.49 NA objective. Images were acquired using MetaMorph.software. SLBs were prepared for cellular activation as described above. Jurkat T cells expressing LAT-mCherry, Nck-sfGFP, and LifeAct-BFP were activated on the SLB for 5 min to allow the IS to form. Confocal slices were then captured with a 0.25 µm step-size. 3-dimensional images were reconstructed using Matlab and the position of the dense actin ring (LifeAct-BFP) and membrane (as indicated by LAT-mCherry fluorescence) were measured and analyzed for spatial orientation using Matlab.

## Image data analysis and display

### Drift correction for In Vitro movies

Due to imaging multiple time points at multiple stage positions for all in vitro reconstitution assay imaging, movies were subject to drift artifacts. To correct for drift, we aligned the frames in the pLAT channel by the maximum pixel-level cross correlation between adjacent frames. We then applied the shift to all channels to ensure identical corrections.

There were no drift artifacts from cellular imaging data because a single stage position was used.

### pLAT condensates in vitro within No actin/Actin/Actomyosin Networks at Steady State

#### Detection of pLAT condensates

To detect pLAT condensates, we first pre-processed the pLAT condensate images by (i) subtracting inhomogeneous background, where the background image was estimated by filtering the pLAT image with a large Gaussian kernel ($\sigma$ = 10 pixels), and (ii) suppressing noise by filtering the background subtracted image with a small Gaussian kernel ($\sigma$ = 1). Next we detected pLAT condensates using a combination of local maxima detection (*Jaqaman et al., 2008*) to handle diffraction-limited condensates, which tended to be dim, and intensity-based segmentation (Otsu threshold) to handle larger condensates, which tended to be brighter (*Vega and Jaqaman, 2019*). Applying these two

algorithms resulted in three detection scenarios: (1) A segmented region containing one local maximum within it, taken to represent one condensate. (2) A local maximum not enclosed within a segmented region, representing a dim, diffraction-limited condensate. In this case, a circular area approximating the two-dimensional point spread function (PSF $\sigma$ = 74 nm=0.46 pixels,=>circle radius=3$\sigma$ = 222 nm=1.4 pixels) centered at the local maximum was taken in lieu of a segmented condensate area. (3) A segmented region containing multiple local maxima. This scenario could arise from overlapping nearby condensates or from fluctuating intensity within one large condensate (possibly due to incomplete mixing following the fusion of two or more individual condensates). To distinguish between these two cases, we compared the intensity variation along the line connecting each pair of local maxima in a segmented region to the intensity variation between the center and the edge of isolated condensates (scenario 1, that is one local maximum within one condensate). Specifically, for each pair of local maxima in a segmented region, we averaged their peak intensities and then calculated the ratio of this average to the minimum intensity along the line between them. Similarly, for isolated condensates (scenario 1), we calculated the ratio of their peak intensity to the minimum intensity at the edge (*Figure 1—figure supplement 1*). This provided us with a reference distribution of peak/minimum ratios for truly separate condensates. In particular, we took the 1st percentile of this distribution as a threshold to distinguish between pairs of local maxima with a segmented region belonging to separate condensates (pair peak/minimum ratio $\geq$threshold) and pairs of local maxima belonging to the same condensate (pair peak/minimum ratio <threshold). If a pair of local maxima was deemed to belong to the same condensate, the local maximum with lower peak intensity was discarded. After eliminating superfluous local maxima with this procedure, we finalized the condensate segmentation and center estimation by using watershed on the originally segmented region with the remaining local maxima as seeds for the watershed algorithm (*Figure 1—figure supplement 1*).

## Tracking of pLAT condensates

To track the detected pLAT condensates, we employed our previously developed multiple-particle tracking software 'u-track' (*Jaqaman et al., 2008*; *Jaqaman and Danuser, 2008*), using a search radius of 5 pixels, a gap closing time window of 3 frames, and with the possibility of merging and splitting (all other parameters had default values). The search radius and gap closing time window were optimized by inspecting the distributions of frame-to-frame displacements and gap durations output by the software, and by visual inspection of the tracked condensates.

## Motion analysis of pLAT condensates

To characterize the movement of pLAT condensates, we used previously developed moment scaling spectrum (MSS) analysis of the condensate tracks (*Ewers et al., 2005*; *Ferrari et al., 2001*; *Jaqaman et al., 2011*; *Vega et al., 2018*).

## Analysis of actin enrichment at pLAT condensates

To quantify the enrichment of actin at pLAT condensates (*Figure 1C*), we employed our previously developed point-to-continuum colocalization analysis algorithm (*Githaka et al., 2016*; *Jaqaman, 2016*). Briefly, actin enrichment in each movie was defined as the ratio of actin intensity within pLAT condensates to the actin intensity outside condensates, averaged over all condensates, at the last frame of each movie.

## pLAT condensates In Vitro within contractile actomyosin networks

### STICS analysis of pLAT and actomyosin movement

To capture the overall movement of the actomyosin network and pLAT condensates, we first applied SpatioTemporal Image Correlation Spectroscopy (STICS) analysis to each channel separately (*Ashdown et al., 2014*; *Vega and Jaqaman, 2019*). STICS was preferred over particle tracking to analyze the pLAT channel in this assay for two reasons: 1) Experimental conditions (described above) resulted in many pLAT → N-WASP condensates strongly wetting actin filaments (when compared with steady-state experiments), thus hindering accurate detection of individual condensates; and 2) using STICS on both channels allowed direct comparisons of speed and directionality at specific image sub-regions. To perform STICS analysis, we used the STICS Fiji plug-in provided by the Stowers Institute (http://research.stowers.org/imagejplugins/index.html), and the java package Miji

(https://imagej.net/Miji) to call Fiji from MATLAB via a short script. Analysis was performed on image sub-regions of size 16 × 16 pixels (16 pixels = 2.56 μm) with a step size of 8 pixels across the image, and with a temporal correlation shift of 3 frames (i.e. 15 s). This was found to be the smallest shift possible to capture both condensate and actin movement without noise dominating the measurement. To capture movements relevant to contraction, STICS analysis was applied to movies only within the time interval of contraction, which lasted 10–15 frames (i.e. 50–75 s).

### Analysis of pLAT and actomyosin c-movement

After acquiring STICS movement vector fields for the two channels (over the contraction time interval), we compared the vector magnitudes (i.e. speeds) and angles at each image sub-region between the two channels (CODE WILL BE UPLOADED TO GITHIB). The angle distribution (e.g. *Figure 2D*; taken from the *histogram* function and plotted using the *stairs* function) and speed comparison (e.g. *Figure 2C*; plotted using the MATLAB function *histogram2*) for any condition were comprised of all the measurements from all sub-regions of all movies representing that condition. Angle distributions were compared between conditions (or between a condition and its randomized control) via a Kolmogorov-Smirnov (KS) test. For this, due to the hyper-sensitivity of the KS test when the distributions comprise many data points (roughly >1000 data points), we performed the KS test on pairs of subsamples from the distributions (down to 500 data points each), repeated 100 times, and the average p-value of the 100 repeats was reported as the KS test p-value (e.g. in *Figure 2D*).

### Live cell data

### Synapse and cSMAC segmentation

To identify the location of LAT condensates relative to the synapse edge and cSMAC center, we first segmented the synapse and the cSMAC. Both were done in a semi-automated manner. For synapse segmentation, the actin channel (LifeAct – BFP) was used when available for a condition, as this channel provided the clearest synapse edge for segmentation. In conditions where actin was not labeled, the LAT channel was used. Images were first smoothed using a Gaussian kernel (σ = 2 pixels), and then an Otsu threshold was applied to separate the image into two levels, that is synapse and background (using the MATLAB function *multithresh*, which generalizes the Otsu method to determine thresholds for a multimodal distribution). We retained only the synapse at the center of the image for further analysis, identified as the largest thresholded object in the image.

For cSMAC segmentation, the LAT channel was always used. In this case, *multithresh* was tasked to determine two thresholds that would separate the LAT image into three levels. The cSMAC was taken as the largest segmented area at the highest intensity level within the segmented synapse. For early frames in which the cSMAC had not yet formed, instead of segmenting the cSMAC we used the point that would eventually become the cSMAC center as an alternative reference. Specifically, we applied the above process but on the average of all the time-lapse frames, and through this the center of the eventual cSMAC was determined and used in those early frames (area = 0 in this case since cSMAC had not formed yet).

We then visually inspected all segmentation results (synapse and cSMAC) and in some cases manually refined the segmentation using in-house software (*Vega and Jaqaman, 2019*).

### Detection and tracking of LAT condensates

Due to the majority of condensates being diffraction limited and a lower SNR in our cellular data, thresholding as described above in 2a for in vitro condensates was not appropriate for cellular analysis, as it lacked the sensitivity to detect individual condensates in cells. Instead, we detected pLAT condensates solely using local maxima detection and Gaussian mixture-model fitting for sub-pixel localization (*Jaqaman et al., 2008*).

After detection, we tracked the LAT condensates in the same way and with the same parameters described in section 2b for in vitro condensates.

### Defining a normalized radial position between synapse edge and cSMAC center

Because synapse and cSMAC size differed between cells, and additionally synapses were not circular, we sought a unitless, normalized measure of position that allowed us to pool measurements

between cells, and even from different parts of the same cell. To this end, for any point in the synapse, we drew a straight line from the cSMAC center to the synapse edge going through that point, and then defined the point's normalized radial position as the ratio of the distance between the point and the cSMAC center to the total line length. With this, the normalized radial position ranged from zero at the cSMAC center to one at the synapse edge. All LAT condensate and actin trends were then measured vs. this normalized radial position (e.g. *Figure 4C*).

## LAT condensate composition analysis

To analyze the composition of LAT condensates as they moved from the synapse edge toward the cSMAC, we selected condensates based on their track duration, track geometry, track start and end position, and initial Nck/Grb2 content, as explained next in detail (*Vega and Jaqaman, 2019*):

1. Track duration:
   Only tracks lasting a minimum of 5 frames were used for any analysis, to obtain enough information per track.
2. Track geometry:
   We reasoned that LAT condensates moving toward the cSMAC should be overall linear (asymmetric). Therefore, for this analysis we selected tracks that were approximately linear, as assessed by measuring the degree of anisotropy of the scatter of condensate positions along the track (*Huet et al., 2006*; *Jaqaman et al., 2008*).
3. Track start and end position:
   In addition to being approximately linear, tracks used for composition analysis were filtered by their start and end positions. To this end, we used the rise and fall of actin intensity as one traversed the synapse from the edge to the cSMAC to define a position threshold (described next), such that LAT condensate tracks used for composition analysis had to start between the synapse edge and this threshold, and had to end between this threshold and the cSMAC center.
   To define this position threshold, we took for every cell at each frame eight actin intensity profiles from the synapse edge to the cSMAC center, using straight lines with a 45° angle between each line and the next (*Figure 4—figure supplement 2*) (using the MATLAB function *improfile*). Pooling the intensity profiles from all time points for all cells showed that the actin intensity peak had a median normalized radial position of ~0.6 (*Figure 4—figure supplement 2*). Therefore, we used this as the threshold dividing the start and end position of tracks used for composition analysis.
4. Initial Nck/Grb2 content:
   As explained in the next paragraph, for composition analysis the amount of Nck/Grb2 in a LAT condensate at any time point was normalized by its amount when the condensate first appeared (specifically the average of the first three frames to account for fluctuations). To exclude condensates that had low amounts of Nck or Grb2 to begin with, only condensates that contained an average Nck or Grb2 protein intensity (as defined next) in their first three time points greater than the average standard deviation of the background intensity in the first three time points were included in the analysis (on average 88% of the tracks surviving conditions 1–3 above).

To quantify protein (i.e. Nck, Grb2, or LAT) content in a condensate, we subtracted local background from the protein intensity inside the condensate and then took the average background subtracted intensity as a measure of protein content. To estimate local background, we determined which condensates were in each other's proximity (referred to as 'condensate aggregates') and then calculated the average and standard deviation of intensity in a two pixel thick perimeter around each condensate aggregate (thus all condensates within one aggregate had the same local background level). For composition analysis over the lifetime of each track, the protein content at each of its time points was normalized by its average protein content at its first three time points. The normalized protein content was then pooled for all condensates based on their frame-by-frame normalized radial position, specifically 0.9–1 (closest to the synapse edge), 0.8–0.9, 0.7–0.8, etc. The ratio of pooled normalized protein content (either Grb2: LAT or Nck: LAT) was plotted as a minimal boxplot showing only the median and notches, indicating the 95% confidence interval of each median (e.g. *Figure 4C*).

## Measuring deviation from straight path for condensate tracks

In order to measure whether condensates retaining a molecular clutch via Grb2$_{Basic}$ displayed aberrant movement as they traversed the two actin networks, we measured the deviation of each condensate track from a straight path towards the cSMAC (*Vega and Jaqaman, 2019*). Before measuring track deviation from a straight path, it was necessary to remove the following two parts of each condensate track, as these parts were irrelevant to the question of interest: (1) the initial time points of a track where the condensate was moving along the synapse edge instead of toward the cSMAC; and (2) the late time points of a track where the condensate had entered the cSMAC (*Figure 4—figure supplement 2*).

To identify the initial part of a track, that is condensate movement along the synapse edge rather than toward the cSMAC, we measured both the frame-to-frame change in distance between the condensate and the cSMAC center and the frame-to-frame change in angle between consecutive displacements taken by the condensate. The rationale was that before a condensate began to move towards the cSMAC, the change in distance should be low, while the change in angle should be high. On the other hand, once a condensate began to move towards the center, the change in distance should be high, while the change in angle should be low. Therefore, to detect the transition between moving along the synapse edge and moving toward the cSMAC, we took the ratio of the change in distance to change in angle at every track time point and chose the first time point with a value in the top 10% of all ratios within the track as the transition point. In the majority of cases (>90% of tracks), the condensate position at this time point was before the actin position threshold previously discussed. However, in some cases, primarily in the case of shorter tracks, the condensate position at the transition time point chosen in this manner was after the actin position threshold. In this case, the transition time point was taken as the first time point with a change in distance-to-change in angle ratio in the top 25%. Manual inspection verified that with this strategy all transition time points were indeed before the actin position threshold.

After the initial and final track parts were removed, the remaining track segment was transformed such that the beginning and end were set to zero on the y-axis. Thus any non-zero y-position along the track could be directly measured as a deviation from a straight path. Additionally, tracks were flipped if necessary so that the majority of their deviations were in the positive direction. The entire deviation distribution from all time points of all tracks was then taken from the *histogram* function and plotted using the *stairs* function (*Figure 5F*). Example tracks were plotted using the function *scatter* and colored to represent the deviation value at each position (*Figure 5E*).

## Measuring extent of photobleaching in live cell data

In order to determine whether photobleaching might contribute to measured changes in condensate composition, we measured the change in cell background intensity over time. The mean intensity within the segmented synapse, but outside the segmented cSMAC and detected condensate areas, was taken for each cell for every frame. All mean intensity measurements were normalized by the mean intensity of the first frame for each cell. The ratio of pooled normalized mean intensity (either Grb2: LAT or Nck: LAT) was plotted as a minimal boxplot showing only the median and notches, indicating the 95% confidence interval of each median (*Figure 4—figure supplement 3*).

## Database Information

Data are available in the BioStudies database (http://www.ebi.ac.uk/biostudies) under accession number S-BIAD6. Image data are available in the Image Data Resource (IDR) (https://idr.openmicroscopy.org) under accession number idr0055.

## Acknowledgements

We thank L Rice, J Hammer III, and our fellow HCIA Summer Institute scientists for stimulating discussions about this study. This work was supported by a Howard Hughes Medical Institute Collaborative Innovation Award, the Welch Foundation (I-1544 to MKR), a JC Bose Fellowship from the Department of Science and Technology, government of India (SM), a Margadarshi Fellowship from the Wellcome Trust – Department of Biotechnology, India Alliance (IA/M/15/1/502018 to SM), NIH (R01 GM100160 to TT) (R35 GM119619 to KJ), a CPRIT Recruitment Award (R1216 to KJ), and the UT Southwestern Endowed Scholars Program (KJ). Research in the Rosen lab is supported by the

Howard Hughes Medical Institute. JAD was supported by a National Research Service Award F32 (F32 DK101188). ARV was supported by a CPRIT Training Grant (RP140110, PI: Michael White). DVK was supported by fellowships of the AXA Research Fund and the National Centre for Biological Sciences, Tata Institute for Fundamental Research. XS was supported by a Cancer Research Institute Irvington postdoctoral fellowship.

## Additional information

### Funding

| Funder | Grant reference number | Author |
| --- | --- | --- |
| Howard Hughes Medical Institute | | Ronald D Vale<br>Michael K Rosen |
| Welch Foundation | I-1544 | Michael K Rosen |
| Department of Science and Technology, Ministry of Science and Technology | J C Bose Fellowship | Satyajit Mayor |
| Wellcome Trust/DBT India Alliance | Margadarshi Fellowship (IA/M/15/1/502018) | Satyajit Mayor |
| National Institutes of Health | R01 GM100160 | Tomomi Tani |
| UT Southwestern | Endowed Scholars Program | Khuloud Jaqaman |
| National Institutes of Health | National Research Service Award F32 (F32 DK101188) | Jonathon A Ditlev |
| Cancer Prevention and Research Institute of Texas | Training Grant RP140110 PI: Michael White | Anthony R Vega |
| National Centre for Biological Sciences | | Darius Vasco Köster |
| Cancer Research Institute | | Xiaolei Su |
| National Institutes of Health | R35 GM119619 | Khuloud Jaqaman |
| National Institutes of Health | F32 DK101188 | Jonathon A Ditlev |
| Cancer Prevention and Research Institute of Texas | R1216 | Khuloud Jaqaman |
| AXA Research Fund | | Darius Vasco Köster |

The funders had no role in study design, data collection and interpretation, or the decision to submit the work for publication.

### Author contributions

Jonathon A Ditlev, Conceptualization, Data curation, Formal analysis, Investigation, Methodology, Writing—original draft, Project administration, Writing—review and editing; Anthony R Vega, Conceptualization, Data curation, Software, Formal analysis, Investigation, Methodology, Writing—review and editing; Darius Vasco Köster, Conceptualization, Data curation, Formal analysis, Investigation, Methodology, Writing—review and editing; Xiaolei Su, Conceptualization, Investigation, Writing—review and editing; Tomomi Tani, Data curation, Formal analysis, Investigation, Methodology, Writing—review and editing; Ashley M Lakoduk, Investigation; Ronald D Vale, Satyajit Mayor, Michael K Rosen, Conceptualization, Supervision, Funding acquisition, Methodology, Project administration, Writing—review and editing; Khuloud Jaqaman, Conceptualization, Software, Supervision, Funding acquisition, Methodology, Project administration, Writing—review and editing

### Author ORCIDs

Jonathon A Ditlev https://orcid.org/0000-0001-8287-7700
Anthony R Vega https://orcid.org/0000-0002-4464-6482

Darius Vasco Köster (iD) https://orcid.org/0000-0001-8530-5476
Ronald D Vale (iD) https://orcid.org/0000-0003-3460-2758
Satyajit Mayor (iD) https://orcid.org/0000-0001-9842-6963
Khuloud Jaqaman (iD) https://orcid.org/0000-0003-3471-1911
Michael K Rosen (iD) https://orcid.org/0000-0002-0775-7917

### Decision letter and Author response

Decision letter https://doi.org/10.7554/eLife.42695.060
Author response https://doi.org/10.7554/eLife.42695.061

## Additional files

### Supplementary files

• Supplementary file 1. Key Resource Table. Table containing information about bacterial strains, cell lines, antibodies, recombinant DNA, peptide/recombinant protein, chemical compounds, software, reagents, and columns used for this study.
DOI: https://doi.org/10.7554/eLife.42695.053

• Transparent reporting form
DOI: https://doi.org/10.7554/eLife.42695.054

### Data availability

Data are available in the BioStudies database (http://www.ebi.ac.uk/biostudies) under accession number S-BIAD6. Image data are available in the Image Data Resource (IDR) (https://idr.openmicroscopy.org) under accession number idr0055. Condensate analysis code is available on GitHub at https://github.com/kjaqaman/CondensateAnalysis (copy archived at https://github.com/elifesciences-publications/CondensateAnalysis). Colocalization analysis code is available on GitHub at https://github.com/kjaqaman/ColocPt2Cont. Cluster tracking analysis code is available on GitHub at https://github.com/DanuserLab/u-track. Polarization microscopy analysis code is available on GitHub at https://github.com/mattersoflight/Instantaneous-PolScope.

The following datasets were generated:

| Author(s) | Year | Dataset title | Dataset URL | Database and Identifier |
|---|---|---|---|---|
| Ditlev JA, Vega AR, Köster DV, Su X, Tani T, Lakoduk AM, Vale RD, Mayor S, Jaqaman K, Rosen MK | 2019 | A Composition-Dependent Molecular Clutch Between T Cell Signaling Condensates and Actin | https://wwwdev.ebi.ac.uk/biostudies/studies/S-BIAD1 | EMBL Biostudies, S-BIAD6 |
| Ditlev JA, Vega AR, Köster DV, Su X, Tani T, Lakoduk AM, Vale RD, Mayor S, Jaqaman K, Rosen MK | 2019 | Imaging data from A composition-dependent molecular clutch between T cell signaling condensates and actin | https://idr.openmicroscopy.org/tissue/search/?query=Name:idr0055 | IDR Open Microscopy, idr0055 |

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
