## [Decision Letter]

Thank you for submitting your article "A Composition-Dependent Molecular Clutch Between T Cell Signaling Condensates and Actin" for consideration by *eLife*. Your article has been reviewed by three peer reviewers, and the evaluation has been overseen by a Reviewing Editor and Arup Chakraborty as the Senior Editor. The following individual involved in review of your submission has agreed to reveal his identity: Michael L Dustin (Reviewer #2).

As you can see, the reviewers have discussed the reviews with one another and the Reviewing Editor has drafted this decision to help you prepare a revised submission. They like the paper, but request some further clarification.

This manuscript aims to define the mechanism by which T cell receptor (TCR) phase-separated microclusters are coupled to actin retrograde flow to establish spatial signaling domains in the immune synapse between a T cell and antigen presenting cell. The authors use in vitro reconstitutions with purified components of the clusters including phospho-LAT, Grb2, Sos1, phospho-Slp-76, and an N-WASP/Wip chimera, together with actomyosin networks coupled to supported lipid bilayers to mimic the T-cell cortex and plasma membrane, and compare this to the dynamics of TCR microclusters in Jurkat T-cells spreading on ligand-coated supported lipid bilayers.

Using the in vitro reconstitution system, the authors discovered a new mechanism that mediates TCR microcluster interactions with either stationary or myosin-driven actin filaments through Nck and N-WASP/Wip. Association of in vitro assembled clusters with actin filaments coupled to supported lipid bilayers involved basic regions of Nck and N-WASP/Wip, and was independent of the WH2 motifs which mediate Arp2/3 activation. Co-sedimentation assays suggested that binding through this motif was low affinity for N-WASP/Wip (although affinity was not directly determined) and absent for Nck, suggesting that the Nck-actin interaction may be dependent on association with other proteins in the clusters. Changing the net charge of the basic domains affected the association of microclusters with actin filaments, suggesting an electrostatic interaction mechanism. These experiments thus establish a novel basic actin-binding motif whose affinity for actin could possibly be regulated by association with phase-separated microclusters. The authors also performed experiments in Jurkat T cells interacting with ligand-coated supported lipid bilayers showing that the composition of TCR microclusters changed during their centripetal movement, with Nck dissociating as microclusters approached the synapse center, correlating with the location in the synapse where actin filaments switch from radially to concentrically oriented. The dynamics of WASP was not examined. They showed that addition of a basic domain to Grb2 is sufficient for mediating the recruitment of phospho-LAT clusters to actin filaments on supported lipid bilayers in vitro and caused LAT-containing microclusters to move less directionally towards the center of the synapse in cells.

Based on these results, the authors conclude that Nck and WASP act to couple ligand-engaged TCR microclusters to actin filaments undergoing retrograde flow in the immune synapse, likening them to the proteins that link ligand-engaged integrins to actin in focal adhesions in what has been termed a "molecular clutch." The biochemical characterization of the molecular determinants of Nck and N-WASP association with actin filaments provides novel and valuable insight into how TCR microsclusters interact with the actin cytoskeleton at the IS.

The quality of the work is outstanding and the additional of + charge to the condensates increases interactions with F-actin in region dominated by the actinomyosin arcs. So they have generated a clear hypothesis and testing it in vitro and in vivo. The study is important in that it reveals a new way for signaling complexes to interface with F-actin cytoskeleton.

However as it stands the reviewers feel that neither the in vivo nor the in vitro work has been taken quite to the stage where it presents a comprehensive picture. We see two ways to improve this, either through the in vivo or the in vitro work.

For the in vivo work, the reviewers have the concern that the cells falls short of demonstrating that Nck and WASP constitute a molecular clutch. One idea you could consider is to knock down Slp76, the molecule that recruits Nck (and then WASP) to clusters. If this doesn't kill the cells, the clusters may move aberrantly in the periphery because they'll lack the clutch. Or perhaps you could overexpress a Slp-76 that can't be phosphorylated on the Nck binding sites, which might also make clusters that don't recruit Nck/WASP as well.

Alternatively, you could strengthen some of the biochemistry of the interactions. The actin-binding activity of Nck and N-WASP/Wip and its regulation by microcluster proteins and their phase separation behavior is not as well-characterized as it might be. The affinity of N-WASP/Wip for actin filaments should be determined in co-sedimentation assays, the effects of individual and combinations of microcluster proteins on Nck and N-WASP/Wip binding to actin should be characterized in co-sedimentation assays, and the effects of combinations of microcluster proteins (and their ability to phase separate) on cluster binding to actin should be characterized in microscopic assays.

We think that overall, if you were to tone down your conclusions, this would be very helpful. It is clear that you have shown that there is a clutch in vitro. The in vivo experiments could be better cast to show that they are supporting, but that future work on more precise perturbations will be necessary.

---

## [Author Response]

[…] However as it stands the reviewers feel that neither the in vivo nor the in vitro work has been taken quite to the stage where it presents a comprehensive picture. We see two ways to improve this, either through the in vivo or the in vitro work.

We thank the reviewers for their generally positive assessment of our work. In our revisions we attempted to improve both the biochemical and cell biological components of the work. We were quite successful with the biochemical work, but our cellular experiments failed. We hope the reviewers feel that the improved biochemistry is sufficient for publication in *eLife*.

For the in vivo work, the reviewers have the concern that the cells falls short of demonstrating that Nck and WASP constitute a molecular clutch. One idea you could consider is to knock down Slp76, the molecule that recruits Nck (and then WASP) to clusters. If this doesn't kill the cells, the clusters may move aberrantly in the periphery because they'll lack the clutch. Or perhaps you could overexpress a Slp-76 that can't be phosphorylated on the Nck binding sites, which might also make clusters that don't recruit Nck/WASP as well.

These are good suggestions, and we worked hard to obtain such data. In the end, however, we were unsuccessful. We obtained mutagenized Jurkat T cells from Dr Art Weiss in which SLP-76 had been knocked out (Cell line J14). We attempted to create four stable cell lines using lentivirus transduction: 1) LAT-BFP, SLP-76-sfGFP, mCherry-Actin; 2) LAT-BFP, SLP-76-YtoF mutation-sf-GFP (this mutant should not recruit Nck), mCherry-Actin; 3) Nck-mCherry, SLP-76-sfGFP; 4) Nck-mCherry, SLP-76-YtoF mutation-sf-GFP (this mutant should not recruit Nck). Cell lines 1 and 2 were to be used to track LAT condensates with the ability to recruit Nck via phospho-SLP-76 (Cell line 1) and compare movement with condensates that would be unable to recruit Nck because of mutations to the SLP-76-sfGFP transgene (Cell line 2). Cell lines 3 and 4 were intended to be controls so that we could determine whether Nck was recruited to SLP-76 knockout cells re-expressing either wild-type SLP-76 (Cell line 3) or mutant SLP-76 (Cell line 4). Unfortunately, expressing these combinations of transgenes either killed the J14 cells, or impaired the cells so badly that they were unable to form immune synapses with supported lipid bilayers.

Because of the failure of these experiments, we have toned down our conclusions in the manuscript. At various places in the text, beginning with the Abstract, we now say that we have shown that Nck and WASP form a clutch between LAT condensates and actin filaments in vitro, and that these data plus our cellular data suggest a model for condensate movement across the immunological synapse.

Alternatively, you could strengthen some of the biochemistry of the interactions. The actin-binding activity of Nck and N-WASP/Wip and its regulation by microcluster proteins and their phase separation behavior is not as well-characterized as it might be. The affinity of N-WASP/Wip for actin filaments should be determined in co-sedimentation assays, the effects of individual and combinations of microcluster proteins on Nck and N-WASP/Wip binding to actin should be characterized in co-sedimentation assays, and the effects of combinations of microcluster proteins (and their ability to phase separate) on cluster binding to actin should be characterized in microscopic assays.

We performed a series of new experiments to address these comments, and they have significantly improved the paper. First, we performed co-sedimentation assays with increasing concentrations of actin to quantify the affinity of Nck and N-WASP for the filaments. These data are shown in Figure 3—figure supplement 3 and have been described in the third paragraph of the subsection “Phase separation on membranes enhances interactions between Nck/N-WASP and actin filaments”. Nck alone (2 µM) did not bind sufficiently tightly to actin filaments (between 2 and 8 µM) to measure the affinity of the interaction. N-WASP also did not bind well enough to afford a multi-point fit of the data for different actin filament concentrations. But based on the ratio of N-WASP in the supernatant and pelleted with actin filaments for 7.9 µM filaments, we could estimate a K_D_ of 25 µM. We also performed analogous experiments with a fixed concentration of actin filaments (2 µM) and increasing numbers of condensate components (all at 1 µM, to avoid phase separation in solution), to determine whether these ligands might increase the affinity of N-WASP for actin filaments. As shown in Figure 3—figure supplement 4, identical amounts of N-WASP co-sedimented with actin filaments in all combinations, including pLAT/Grb2/pSLP-76/Nck/N-WASP, which represents the full set of proteins in membrane clusters. Thus, increasing the number of N-WASP binding partners did not increase the affinity of N-WASP for actin filaments. We note that the N-WASP ligands do co-sediment together in small amounts, indicating that they are binding N-WASP in solution (and associating with actin filaments through N-WASP). But these interactions do not have a measurable effect on the affinity of N-WASP for actin filaments. Thus, in solution, N-WASP, either alone or with its collection of direct and indirect ligands, binds actin filaments only weakly.

In contrast, quantitative experiments on membrane associated N-WASP clusters showed a much higher affinity for actin filaments. We performed microscopic binding assays using our supported lipid bilayer experimental system to determine the affinity of actin filaments for LAT condensates with compositions of pLAT → Nck and pLAT → NWASP. The results from these experiments are in Figure 3—figure supplement 2 and are described in the second paragraph of the subsection “Phase separation on membranes enhances interactions between Nck/N-WASP and actin filaments”. Briefly, affinities were calculated from assays in which condensates were formed on supported lipid bilayers in the presence of increasing concentrations of rhodamine-labeled actin filaments in the solution, and the fluorescence of actin recruited to the bilayer was measured by TIRF microscopy. Fitting the data to a Hill equation yielded effective K_D_ values for pLAT → Nck and pLAT → NWASP condensates binding to actin of 410 and 280 nM, respectively, with similar Hill coefficients of 3.2 and 3.6, respectively. These data and the solution binding data above yield the interesting conclusion that Nck and N-WASP bind actin filaments with appreciably higher affinity when arrayed on membranes than when dispersed in solution.

In a final set of biochemical experiments, we also purified full-length human WASP, which is the major WASP family member in T cells, and performed both “Steady-State” and “Contraction” assays with pLAT → WASP condensates. We added the relevant TIRF microscopy images and analysis to Figures 1 and 2, and included a new Figure 3—figure supplement 1 in which we compared actin binding to pLAT → Nck, pLAT → NWASP, or pLAT → WASP condensates. In all of these assays, WASP behaves intermediate between Nck and N-WASP in terms of interactions with actin, consistent with the fact that the basic region of WASP has fewer positively charged residues than that of N-WASP. The text describing both “Steady-State” and “Contraction” assays has been adjusted to include our new WASP data. The method for purification of full-length human WASP, information regarding the protein, and relevant experimental details have also been added to the Materials and methods, Table 1, and Table 2, respectively, of the manuscript.

Thus, individually weak interactions can be converted into high affinity collective interactions when signaling molecules are clustered on membranes. We believe that this is a very important emergent property of the generation of the condensate with the generation of a relevant composition (of Nck and N-WASP or WASP).